# Nanophysiology approach reveals diversity in calcium microdomains across zebrafish retinal bipolar ribbon synapses

Nirujan Rameshkumar[1†], Abhishek P Shrestha[1†‡], Johane M Boff[1†], Mrinalini Hoon[2,3,4], Victor Matveev[5], David Zenisek[6,7], Thirumalini Vaithianathan[1,8*]

[1]Department of Pharmacology, Addiction Science, and Toxicology, The University of Tennessee Health Science Center, Memphis, United States; [2]Department of Neuroscience, University of Wisconsin, Madison, United States; [3]McPherson Eye Research Institute, University of Wisconsin, Madison, United States; [4]Department of Ophthalmology and Visual Sciences, University of Wisconsin, Madison, United States; [5]Department of Mathematical Sciences, New Jersey Institute of Technology, Newark, United States; [6]Department of Cellular and Molecular Physiology, Yale University School of Medicine, New Haven, United States; [7]Department of Ophthalmology and Visual Sciences, Yale University School of Medicine, New Haven, United States; [8]Department of Ophthalmology, Hamilton Eye Institute, University of Tennessee Health Science Center, Memphis, United States

*For correspondence: tvaithia@uthsc.edu

†These authors contributed equally to this work

Present address: ‡Department of Surgery, College of Medicine, University of Florida, Florida, United States

Competing interest: The authors declare that no competing interests exist.

## eLife Assessment

The study introduces new tools for measuring the intracellular calcium concentration close to transmitter release sites, which may be relevant for synaptic vesicle fusion and replenishment. This approach yields **important** new information about the spatial and temporal profile of calcium concentrations near the site of entry at the plasma membrane. This experimental work is complemented by a coherent, open-source, computational model that successfully describes changes in calcium domains. The conclusions are **solid** and well supported by the data.

**Abstract** Rapid and high local calcium ($Ca^{2+}$) signals are essential for triggering neurotransmitter release from presynaptic terminals. In specialized bipolar ribbon synapses of the retina, these local $Ca^{2+}$ signals control multiple processes, including the priming, docking, and translocation of vesicles on the ribbon before exocytosis, endocytosis, and the replenishment of release-ready vesicles to the fusion sites for sustained neurotransmission. However, our knowledge about $Ca^{2+}$ signals along the axis of the ribbon active zone is limited. Here, we used fast confocal quantitative dual-color ratiometric line-scan imaging of a fluorescently labeled ribbon binding peptide and $Ca^{2+}$ indicators to monitor the spatial and temporal aspects of $Ca^{2+}$ transients of individual ribbon active zones in zebrafish retinal rod bipolar cells (RBCs). We observed that a $Ca^{2+}$ transient elicited a much greater fluorescence amplitude when the $Ca^{2+}$ indicator was conjugated to a ribeye-binding peptide than when using a soluble $Ca^{2+}$ indicator, and the estimated $Ca^{2+}$ levels at the ribbon active zone exceeded 26 μM in response to a 10 millisecond stimulus, as measured by a ribbon-bound low-affinity $Ca^{2+}$ indicator. Our quantitative modeling of $Ca^{2+}$ diffusion and buffering is consistent with this estimate and provides a detailed view of the spatiotemporal $[Ca^{2+}]$ dynamics near the ribbon. Importantly, our data demonstrates that the local $Ca^{2+}$ levels may vary between ribbons of different RBCs and within the same cells. The variation in local $Ca^{2+}$ signals is found to correlate with ribbon size and active zone extent. Our serial electron microscopy results provide new information

about the heterogeneity in ribbon size, shape, and area of the ribbon in contact with the plasma membrane.

## Introduction

Sensory synapses in the retina rely on the proper function of a specialized organelle, the synaptic ribbon (*Matthews and Fuchs, 2010*; *Moser et al., 2020*; *Regus-Leidig and Brandstätter, 2012*; *Snellman et al., 2011*; *Zanazzi and Matthews, 2009*). Retinal bipolar cells serve as the major conduit for transmitting visual information across the vertebrate retina. RBCs can release brief bursts of neurotransmitters to signal a change in contrast or sustain the continuous release of neurotransmitters in a graded manner to provide an analog read-out of luminance (*Oesch and Diamond, 2011*). To maintain this ability, the RBCs must exert dynamic control over neurotransmitter release rate and facilitate efficient recruitment of release-ready vesicles to fusion sites near the synaptic ribbon. It is established that the elevation of presynaptic $Ca^{2+}$ in RBCs regulates both dynamic changes in the release rate and accelerates the rate of vesicle replacement (*Oesch and Diamond, 2011*; *Burrone et al., 2002*; *Coggins and Zenisek, 2009*; *Graydon et al., 2011*; *Jarsky et al., 2010*; *Mennerick and Matthews, 1996*; *Singer and Diamond, 2003*; *Snellman et al., 2009*; *Von Gersdorff and Mathews, 1994*). Of note, nanodomain calcium signals refer to highly localized, steep calcium concentration gradients that occur within tens to a couple of hundred nanometers of an open calcium channel. They are typically involved in triggering fast, synchronous neurotransmitter release, especially in synapses where $Ca^{2+}$ sensors are closely coupled to the channels (*Augustine et al., 1991*; *Augustine, 1990*; *Adler et al., 1991*; *Eggermann et al., 2011*). Microdomain $Ca^{2+}$ signals extend over a larger spatial range and arise when multiple calcium channels open in proximity, leading to overlapping calcium plumes. These domains result in slower, more diffuse $Ca^{2+}$ elevations, often associated with modulatory functions or asynchronous release, where the $Ca^{2+}$ sensor is located farther from the channel cluster (*Eggermann et al., 2011*; *Kittel et al., 2006*; *Meinrenken et al., 2003*). However, the spatiotemporal properties of the $Ca^{2+}$ signals that control neurotransmitter release and the molecular entities that regulate the interplay between $Ca^{2+}$ signals and synaptic vesicle dynamics to sustain kinetically distinct neurotransmitter release components remain poorly understood. Here, we begin to address this lack of knowledge by measuring local $Ca^{2+}$ signals at positions along the synaptic ribbon at different distances from the active zone in retinal RBCs.

Resolving local $Ca^{2+}$ signals is technically challenging as it requires information about the spatiotemporal properties of $Ca^{2+}$ signals specific to ribbon sites. Previous studies of $Ca^{2+}$ dynamics in goldfish and mammalian RBCs focused primarily on the terminal as a whole and used quantitative methods to examine bulk $Ca^{2+}$ levels, which are significantly lower and slower than those occurring at the active zone (*Burrone and Lagnado, 2000*; *Kobayashi and Tachibana, 1995*; *Wan et al., 2012*; *Zenisek and Matthews, 2000*). Our previous studies estimated that the $Ca^{2+}$ signals at a single ribbon active zone in zebrafish RBCs likely exceed sub-micromolar concentration levels within 3 milliseconds after the opening of voltage-gated $Ca^{2+}$ (Cav) channels located beneath the synaptic ribbon (*Vaithianathan and Matthews, 2014*). However, these studies were based on estimates using soluble $Ca^{2+}$ indicators to estimate the $Ca^{2+}$ signals at the plasma membrane, which are free to diffuse and thus cause the spread of the signal. Measurements of $Ca^{2+}$ signals along the axis of the ribbon have not been attempted, even though these signals likely control replenishment and priming of vesicles for slower phases of neurotransmitter release. A practical way to conceptualize the distinction between micro- and nanodomain $Ca^{2+}$ signals is by approximating spatial scales: signals extending from around 0.5 μm and above may be considered microdomains, while those below 0.3–0.4 μm fall within the nanodomain range (*Eggermann et al., 2011*). To determine $Ca^{2+}$ signals along the axis of zebrafish RBC ribbons, we establish a nanophysiology ratiometric approach that measures the spatial and temporal properties of $Ca^{2+}$ transients by targeting the $Ca^{2+}$ indicator to the ribbon via conjugation to a ribbon binding peptide (RBP) (*Francis et al., 2011*). Resolving $Ca^{2+}$ signals in the immediate vicinity of the active zone is not currently possible due to the finite spatiotemporal resolution of optical imaging. Thus, we employ a quantitative model using $Ca^{2+}$ diffusion and buffering to resolve the RBC $Ca^{2+}$ signals associated with the active zone and along the ribbon and provide a detailed description of the spatiotemporal $Ca^{2+}$ dynamics across zebrafish RBC ribbons. Our nanophysiology approach for measuring local $Ca^{2+}$ transients at a single ribbon with high spatiotemporal resolution provides the

first evidence of heterogeneity of Ca²⁺ signals at zebrafish RBC ribbon synapses. Heterogeneity in local Ca²⁺ signals persisted in some ribbons even after the application of high concentrations of exogenous Ca²⁺ buffers, suggesting that the variability in the faster, smaller, and more spatially confined Ca²⁺ microdomains originates from Ca²⁺ influx through synaptic Ca²⁺ channel clusters at the base of the synaptic ribbon. Using serial block-face scanning electron microscopy (SBF-SEM), we found substantial variability in synaptic ribbon size, shape, and particularly the area of the ribbon adjacent to the plasma membrane in the zebrafish bipolar cell terminal. Thus, the observed heterogeneity of the Ca²⁺ microdomain is likely due to variability in the number of Ca²⁺ channels near each ribbon. Since local Ca²⁺ signals control kinetically distinct neurotransmitter release, heterogeneity in local Ca²⁺ signal may alter the rate of vesicle release and allow them to function independently, adding a new mechanism for increasing dynamic range of RBC.

## Results

### Ca²⁺ concentrations at single ribbon locations measured using high- and low-affinity diffusible indicators

Synaptic vesicles in RBCs are distributed among at least four distinct pools based on their fusion kinetics, which are assumed to reflect the average proximity of vesicles to Cav channels and the state of vesicle preparedness for Ca²⁺-triggered fusion (*Matthews and Fuchs, 2010*; *Oesch and Diamond, 2011*; *Coggins and Zenisek, 2009*; *Mennerick and Matthews, 1996*; *Singer and Diamond, 2003*; *Snellman et al., 2009*; *Datta et al., 2017*; *Euler et al., 2014*; *Neves and Lagnado, 1999*; *von Gersdorff and Matthews, 1997*; *Zhou et al., 2006*; see *Figure 1—figure supplement 1*). To visualize and measure Ca²⁺ signals near the RBC synaptic ribbon controlling such kinetically distinct components of neurotransmitter release, we established a quantitative nanophysiology approach shown in *Figure 1* (see also *Materials and methods*). In this approach, both the ribbon location and the spatiotemporal Ca²⁺ signal profile were simultaneously measured by dialyzing the zebrafish RBC terminals with both TAMRA (tetramethyl rhodamine)-labeled RIBEYE-binding peptide (RBP) (*Zenisek et al., 2004*) (to label synaptic ribbons; *Figure 1Aa*) and a high-affinity Ca²⁺ indicator Cal-520 (Cal520HA, effective $K_D$ 795 nM; see *Materials and methods*) using a whole-cell patch pipette placed directly at the cell terminal. A rapid x-t line scan was taken perpendicular to the plasma membrane across a ribbon, extending from the extracellular space to the cytoplasmic region beyond the ribbon (*Figure 1Ab*). The spatial resolution was limited by the point spread function (PSF) of the microscope to approximately 270 nm (*Figure 1—figure supplement 2A*). In our previous studies, line scans have been applied primarily to measure the temporal properties of Ca²⁺ signals (*Vaithianathan and Matthews, 2014*). However, x-t raster plots obtained at a ribbon active zone labeled with RBP allow us to characterize the spatial localization of Ca²⁺ transients relative to the synaptic ribbon and plasma membrane (*Figure 1C–D*) in addition to characterizing its temporal aspects (*Figure 1E–F*). We previously used a similar approach for localizing and tracking single synaptic vesicles before and during fusion at a single ribbon (*Vaithianathan et al., 2016*), and to measure the kinetics of clearance of fused synaptic vesicle membrane in zebrafish RBC (*Vaithianathan et al., 2019*). We found that depolarization-evoked Ca²⁺ influx caused a rapid increase in Ca²⁺ signals at ribbon locations (*Figure 1C*, cyan, white horizontal arrowhead) and increased more slowly and less dramatically in the cytoplasm (*Figure 1C*, cyan, white vertical arrow) in zebrafish RBC. The Sigmoid-Gaussian function fitting of the x-t scans horizontal profile scans (see *Materials and Methods*) shows that the local Ca²⁺ signals increased rapidly during stimuli (*Figure 1D*, cyan line) and then decreased immediately after the end of depolarization (*Figure 1D*, black line), approaching the spatial profile corresponding to the resting Ca²⁺ levels (*Figure 1D*, gray line). As expected, the centroid position of Ca²⁺ signals during depolarization (*Figure 1D*, cyan $x_0$) is closer to the plasma membrane (*Figure 1D*, magenta $x_{1/2}$) than the centroid position of RBP (*Figure 1D* magenta, $x_0$), since Cav channels are located at the membrane (*Vaithianathan and Matthews, 2014*; *Zenisek et al., 2004*; *Beaumont et al., 2005*; *Berntson and Morgans, 2003*; *Issa and Hudspeth, 1994*; *Issa and Hudspeth, 1996*; *Llobet et al., 2003*; *Lv et al., 2012*; *Morgans et al., 2001*; *Neef et al., 2018*; *Raviola and Raviola, 1982*; *Roberts et al., 1990*; *Rodriguez-Contreras and Yamoah, 2001*; *Thoreson et al., 2013*; *Tucker and Fettiplace, 1995*; *Zenisek et al., 2003*).

To characterize the temporal Ca²⁺ profile at different distances from the plasma membrane, we analyzed the x-t line scans along the time-axis at three distinct distances (*Figure 1E–F*) based on

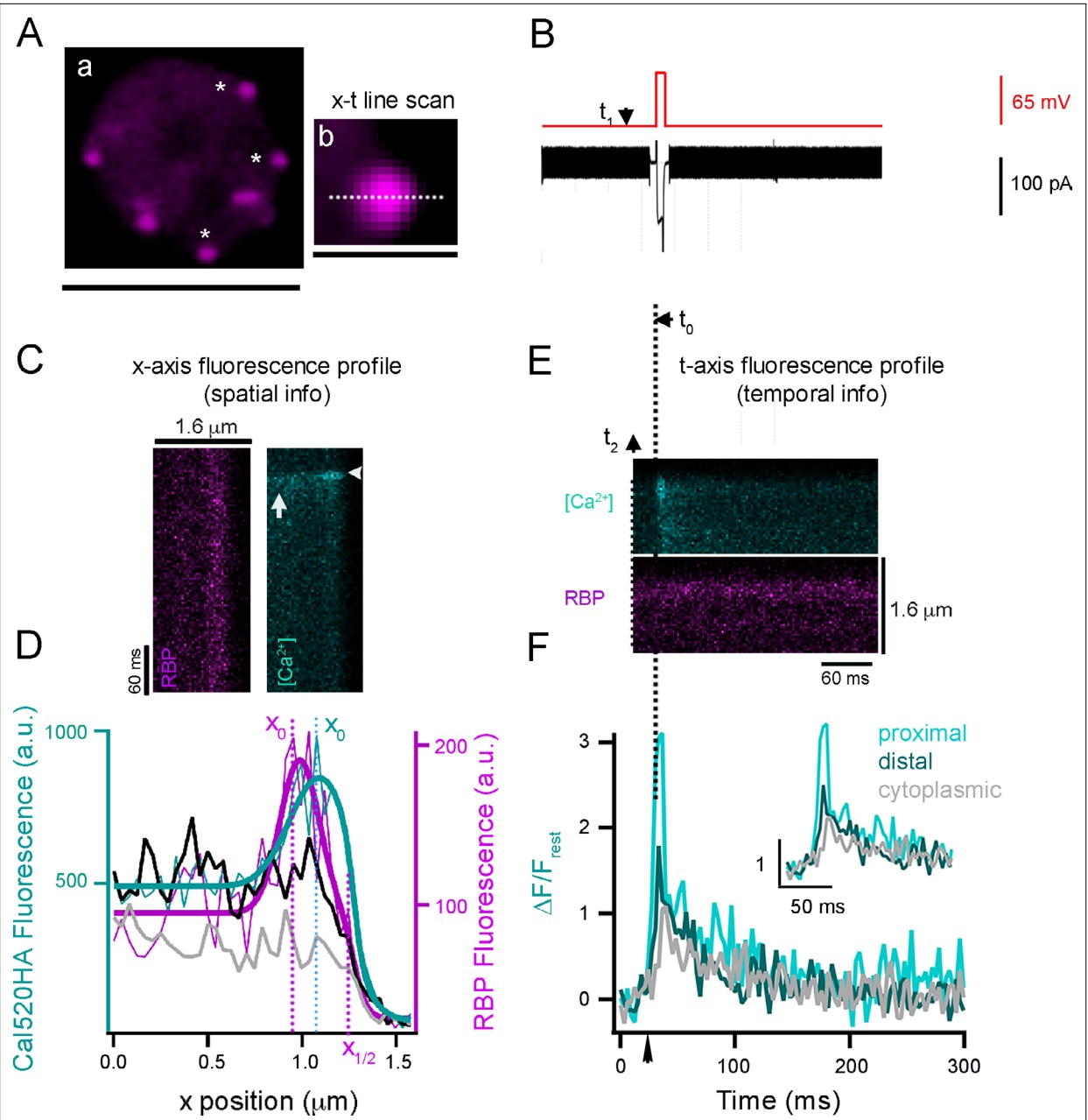

**Figure 1.** Nanophysiology approach unveils spatiotemporal properties of local Ca²⁺ signaling in retinal rod bipolar cells (RBC) terminals. (**A**, Left panel) Single projection from a series of confocal optical sections through a zebrafish RBC synaptic terminal. A synaptic terminal was voltage-clamped using a whole-cell pipette with an internal solution containing TAMRA-RBP (magenta) to label synaptic ribbons (**a**, magenta). Ribbon binding peptide (RBP) fluorescence was concentrated at ribbons and also filled the entire terminal, allowing visualization of the terminal border. Experiments were carried out on ribbons that could be distinguished from adjacent ribbons (white asterisks). Scale, 5 µm. Right panel, Close-up view of a single synaptic ribbon. The outside of the cell is to the right, and x-t scan lines (dotted lines) were positioned perpendicular to the plasma membrane, extending from the intracellular side of the ribbon to the extracellular space. A rapid x-t line scan was taken at a ribbon location perpendicular to the plasma membrane across a ribbon (**b**) with sequential dual laser scanning performed at rates of 1.51 milliseconds per line per channel (3.02 milliseconds per line for both channels). The resulting x-t raster plots were used to measure the fluorescence intensity profiles of RBP (magenta) and the Ca²⁺ transient (cyan; Cal520HA) shown in panels **C** and **E**. Scale, 1.6 µm. (**B**) Voltage-clamp recording of a RBC terminal. Terminals were held at –65 mV and stepped to 0 mV ($t_0$) for 10 ms (red) to evoke a brief Ca²⁺ current (black). A typical experiment began with a voltage command ($V_H$ = –65 mV), and a transistor-transistor logic (TTL) pulse generated by the Patch Master software ($t_1$) triggers image acquisition ($t_2$). $t_0$ is the time of depolarization. (**C**) Illustration of the approach used to obtain the spatial location of Ca²⁺ signals with respect to the ribbon. Example of an x-t raster plot that is oriented to illustrate the x-axis intensity profiles of RBP (magenta) and Ca²⁺ signal (cyan) fluorescence during a brief depolarization. Sequential dual laser scanning was performed at 1.51 milliseconds per line for one channel (3.02 milliseconds per line for both channels). (**D**) Fluorescence intensity profiles along the x-axis for RBP

*Figure 1 continued on next page*

*Figure 1 continued*

(magenta) and Cal520HA before stimuli (gray line), during 10 ms depolarization (cyan line), and after stimuli (black line) depolarization were obtained by averaging three pixels along the time-axis. RBP (magenta) and $Ca^{2+}$ signals during (light cyan) were fit with a Sigmoid-Gaussian function (*Vaithianathan et al., 2016*). The centroid (x-axis position) of the RBP (magenta) and $Ca^{2+}$ signals during (cyan) were taken as the peak of the Gaussian fit ($x_0$). The parameter $x_{1/2}$ (dotted magenta line) from the Sigmoid fit to the RBP fluorescence (magenta trace) was used to estimate the location of the plasma membrane. (E) The x-t raster plot shown was from the same recording as in panel C but re-oriented to demonstrate the t-axis intensity profiles of RBP (magenta) and $Ca^{2+}$ transient (cyan). (F) Spatially averaged Cal520HA fluorescence as a function of time at ribbon proximal (light cyan), distal (dark cyan), and cytoplasmic (gray) locations from the single ribbon shown in D, upper panel.

The online version of this article includes the following figure supplement(s) for figure 1:

**Figure supplement 1.** Ultrastructure of a zebrafish rod bipolar cell (RBC) showing distinct synaptic vesicle pools.

**Figure supplement 2.** $Ca^{2+}$ indicator fluorescence imaging in the rod bipolar cell (RBC) terminal.

the spatial profile of the ribbon described in *Figure 1C–D* (see also *Materials and methods*). We will refer to the corresponding three measurements as *ribbon-proximal*, *ribbon-distal*, and *cytoplasmic* (see *Materials and methods*). We found that at the onset of the stimulus (*Figure 1F*, black arrow), the fluorescence at the location near the ribbon proximal to the membrane (ribbon-proximal) rose more rapidly and to a higher level (*Figure 1F*, light cyan trace) than that at the ribbon-distal and cytoplasmic locations (*Figure 1F*, dark cyan and gray traces, respectively). To quantify the kinetics of ribbon-proximal and distal $Ca^{2+}$ signals, we averaged multiple x-t scans acquired with Cal520HA under the same imaging conditions (*Figure 2A*). The maximum values of trial-averaged $\Delta F/F_{rest}$ (changes in the Cal520HA fluorescence during depolarization normalized to background level before depolarization) were significantly higher at ribbon-proximal locations (nearest the Cav channels) than those at ribbon-distal locations, with mean ± SEM of 1.9±0.3 and 1.5±0.25, respectively (*Figure 2A*; $p<0.001$, N=24). These findings suggest that the spatial resolution of our $Ca^{2+}$ imaging using Cal520HA is sufficient to resolve differences in the smaller $Ca^{2+}$ signals at ribbon proximal and distal locations. However, it should be noted that Cal520HA will be partially saturated at the $Ca^{2+}$ levels expected in $Ca^{2+}$ micro-domains relevant for vesicle exocytosis (*Heidelberger et al., 1994*), affecting both the amplitude and kinetics of the fluorescence signal. Therefore, we repeated the x-t line scan analysis with a lower-affinity soluble $Ca^{2+}$ indicator Cal520LA ($K_D$ 90 µM; *Figure 2B*), allowing us to better define the typical $Ca^{2+}$ signals controlling distinct neurotransmitter release components corresponding to locations proximal and distal relative to the synaptic ribbon (*Figure 2B*). We found that the ribbon-proximal signals detected with Cal520LA (*Figure 2B*, light cyan) showed a sharper decay at the termination of the stimulus (*Figure 2B*), when compared to ribbon-distal signals (*Figure 2B*, dark cyan), as one would expect for nanodomain $Ca^{2+}$ elevations (*Roberts, 1994*; *Naraghi and Neher, 1997*; *Zucker and Fogelson, 1986*). In twenty-one similar experiments, the peak $\Delta F/F_{rest}$ at the membrane after 10 ms depolarization was significantly larger for proximal than distal signals ($\Delta F/F_{rest}$: 3.1 ± d0.4 and 1.9 ± 0.2, respectively $p=0.001$, N=21). As expected, the ratio of proximal to distal signals measured with Cal520LA (1.6) was significantly higher than that measured with Cal520HA (1.3). The decay phase of all fluorescence transients was fit by a sum of two exponential functions, as described in Methods; for the Cal520HA recording (*Figure 2A*) the two decay time constants and their relative magnitudes were similar for the ribbon-proximal and distal recordings, namely 11 ms (60%) and 203 ms (40%) for the ribbon-proximal recording, vs. 16 ms (61%) and 208 ms (39%) for the distal site. For the low-affinity Cal520LA dye (*Figure 2B*), the fluorescence decay components appeared faster due to faster $Ca^{2+}$ unbinding from the dye, at 9.3 ms (69%) and 175 ms (31%) for the ribbon-proximal location, vs. 9 ms (41%) and 135 ms (59%) for the distal location. We note, however, that bi-exponential data fits are known to be highly sensitive to measurement duration and noise. Therefore, precise quantitative conclusions should not be drawn from the best-fit decay time constant values.

## $Ca^{2+}$ concentrations at single ribbon locations measured using ribbon-bound indicators

Although $Ca^{2+}$-sensitive fluorescent chemical dyes have been used previously (*Burrone et al., 2002*; *Francis et al., 2011*; *Beaumont et al., 2005*; *Neef et al., 2018*; *Zenisek et al., 2003*; *Heidelberger et al., 1994*; *Naraghi and Neher, 1997*; *Zucker and Fogelson, 1986*; *Augustine et al., 2003*; *Babai et al., 2010a*; *Beutner et al., 2001*; *Delvendahl et al., 2015*; *Frank et al., 2009*; *Grassmeyer and Thoreson, 2017*; *Grynkiewicz et al., 1985*; *Hosoi et al., 2007*; *Neher and Augustine, 1992*; *Ohn*

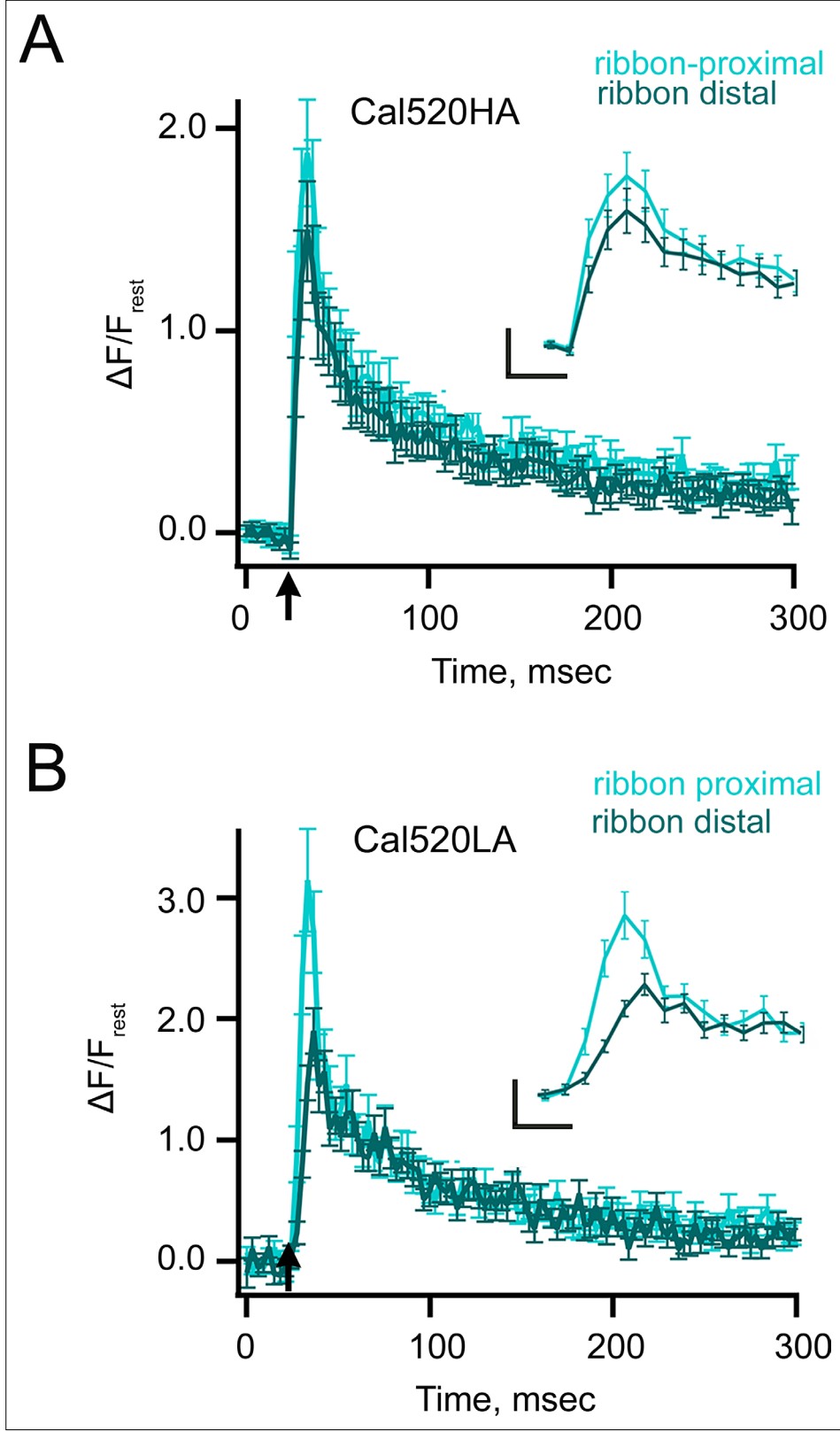

**Figure 2.** Kinetics of Ca²⁺ transients in response to brief stimuli recorded with freely diffusible indicators. (**A**) Spatially averaged Cal520HA fluorescence as a function of time at ribbon proximal (light cyan) and distal (dark cyan) locations from a single ribbon, as shown in *Figure 1* (n=24 ribbons from seven different rod bipolar cells RBCs). The corresponding maximum values of trial-averaged $\Delta F/F_{rest}$ were significantly higher at ribbon-proximal

*Figure 2 continued on next page*

*Figure 2 continued*

locations than those at ribbon-distal locations, with mean ± SEM of 1.9±0.3 and 1.5±0.25, respectively (paired t-test, *p*<0.001, N=24). (**A** inset) The temporal profile of events between 20–60 ms is shown in an expanded view for better visualization. Scale bars: vertical, 0.5 (ΔF/F$_{rest}$); horizontal,10 ms. (**B**) Spatially averaged Cal520LA fluorescence as a function of time at ribbon proximal (light cyan) and distal (dark cyan) locations. Data points show the average intensity (± SEM) in each horizontal row of 5 pixels for three 10 ms depolarizations at distinct ribbon locations (see *Materials and methods* and *Figure 1*). Fluorescence intensity is normalized with respect to the baseline fluorescence before stimulation, and averaged over all pixels (i.e. over space and time). The arrow indicates the onset of the 10 ms depolarizing stimulus. (n=21 ribbons from 4 different RBCs). The peak ΔF/F$_{rest}$ at the membrane after 10 ms depolarization was significantly larger for proximal than distal signals (paired t-test, ΔF/F$_{rest}$: 3.1±0.4 and 1.9±0.2, respectively, p=0.001, N=21). The currents were not significantly different between the Cal520HA and Cal520LA conditions (unpaired t-test, Cal520HA average current = 45.1±4.5 pA; Cal520LA average current = 51.4±4.4 pA; p=0.33). (**B** inset) Temporal profile of events between 20–60 ms was expanded for better visualization. Scale bars: vertical, 0.5 (ΔF/F$_{rest}$); horizontal, 10 ms.

---

*et al., 2016*; *Sakaba and Neher, 2001*), the visualization of signals within smaller domains using freely diffusible Ca$^{2+}$ reporters is limited by the resolution of light microscopy and the spread of the indicator by diffusion (*Schermelleh et al., 2010*). Diffusible Ca$^{2+}$ indicators report space-averaged Ca$^{2+}$ concentrations, and their intracellular diffusion inherently broadens the spatial resolution of Ca$^{2+}$ nanodomains. To partially overcome this problem, we targeted the Ca$^{2+}$ indicators to the ribbon by fusing them to the RBP, as described previously (*Francis et al., 2011*). RBP-conjugated Ca$^{2+}$ indicators Cal520HA-RBP or Cal520LA-RBP were introduced to the RBC terminal together with fluorescently labeled RBP via whole-cell voltage clamp by placing the patch pipette directly at the terminal while imaging the terminal using laser scanning confocal microscopy. For two-color imaging, ribbons were labeled with TAMRA-RBP that did not interfere with Cal520-RBP fluorescence, and both channels were scanned sequentially to prevent possible bleed-through. Under these conditions, we used the spots detected by TAMRA-RBP to define the locus of the synaptic ribbon (*Figure 3A&B*, magenta). We also found punctate regions with both Cal520HA-RBP (*Figure 3A*, cyan) and Cal520LA-RBP (*Figure 3B*, cyan) at the same location, on a dimmer fluorescent background of the synaptic terminal, which correspond to the Ca$^{2+}$ indicator-peptide complexes that are bound and not bound to the ribbon, respectively (*Francis et al., 2011*). The overall changes in ΔF/F$_{rest}$ were averaged over several trials for proximal and distal Ca$^{2+}$ signals measured with Cal520HA-RBP (*Figure 3C*, light cyan vs. dark cyan) and Cal520LA-RBP (*Figure 3D*, light cyan vs. dark cyan). When compared to their distal location counterparts, the Ca$^{2+}$ signals proximal to the membrane showed a sharper decay at the termination of stimuli (*Figure 3C and D*, light vs. dark cyan trace), as expected when comparing nano- and microdomain Ca$^{2+}$ profiles. Our data shows that the amplitude differences between ribbon-proximal and ribbon-distal Ca$^{2+}$ signals were well resolvable using the ribbon-bound Cal520HA-RBP indicator (*Figure 3C*, light vs. dark cyan traces: ΔF/F$_{rest}$ = 3±0.4 vs 1.9±0.3, respectively, *p*=0.001) and Cal520LA-RBP indicator (*Figure 3D*, light vs. dark cyan traces: ΔF/F$_{rest}$ = 5.5±0.9 vs 3.3±0.8, respectively, *p*=0.003). The amplitudes of ribbon-proximal Ca$^{2+}$ signals were higher when measured with Cal520LA-RBP than with Cal520LA-free (*Figure 3E*, Cal520LA-RBP (light cyan) vs. Cal520LA-free (gray): ΔF/F$_{rest}$ = 5.5±0.9, n=30 vs 3.1±0.4, n=21, *p*=0.04) but this was not the case for distal Ca$^{2+}$ signals (*Figure 3F*, Cal520LA-RBP (light cyan) vs. Cal520LA-free (gray): ΔF/F$_{rest}$ = 3.3±0.8, n=30 vs 1.9±0.2, n=21, *p*=0.15). Notably, the Ca$^{2+}$ signals at distal sites measured with Cal520LA-RBP reached their peak amplitude earlier (*Figure 3F* inset, dark cyan) than those measured with Cal520LA-free (*Figure 3F* inset, light gray). The two decay time constants and their relative magnitudes for the Cal520HA-RBP fluorescence (*Figure 3C*) were 11 ms (52.5%) and 182 ms (47.5%) for the ribbon-proximal site, vs. 24 ms (47%) and 192 ms (53%) for the site distal to the ribbon. For the low-affinity ribbon-bound indicator dye (*Figure 3D*), the fluorescence decay components appeared faster, at 5.7 ms (83%) and 129 ms (17%) for the proximal site, vs. 7.5 ms (76%) and 158 ms (24%) for the distal location. Together, these findings suggest that conjugating the Cal520LA indicator to RBP provides a more accurate, promising approach for measuring the distinct local Ca$^{2+}$ signals at ribbon-proximal vs. ribbon-distal locations.

This method also enables ratiometric measurements with RBP-conjugated Ca$^{2+}$ indicator by normalizing its fluorescence to that of RBP-peptides conjugated to Ca$^{2+}$-insensitive fluorophores (TAMRA-RBP) to provide an estimate of Ca$^{2+}$ concentration in RBC synaptic ribbons. We obtained

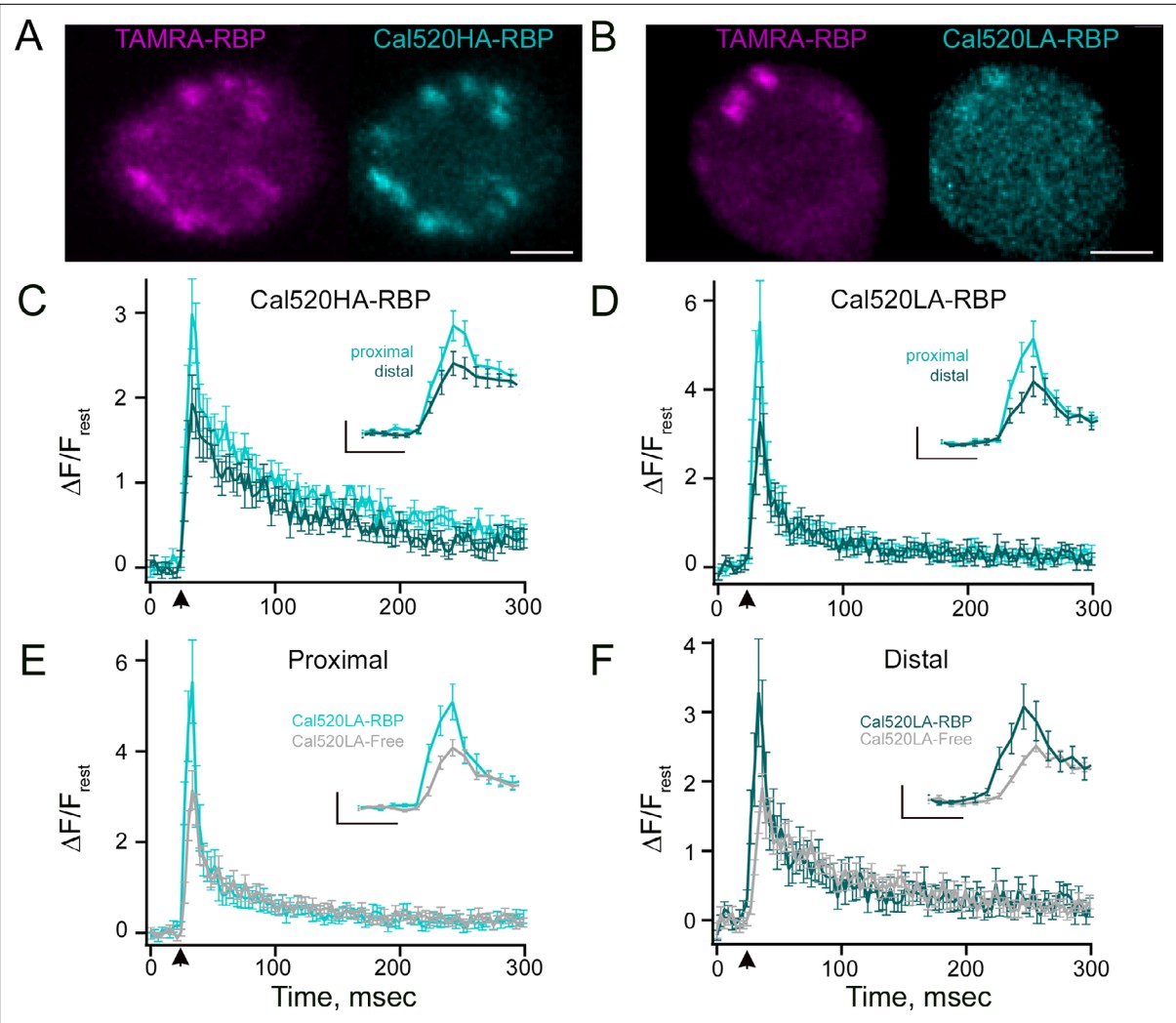

**Figure 3.** Temporal properties of Ca$^{2+}$ transients recorded with free and ribeye-bound Ca$^{2+}$ indicators. (**A–B**) Confocal images of the isolated RBCs that were whole-cell voltage-clamped using an internal solution containing the TAMRA-RBP (magenta) and either (**A**) Cal520HA-RBP (cyan) or (**B**) Cal520LA-RBP (cyan). Note prominent spots in both TAMRA-RBP and ribeye-bound Ca$^{2+}$ indicators (**A**) Cal520HA-RBP (cyan) or (**B**) Cal520LA-RBP, showing the location of the ribbon. Scale bars, 2 μm. (**C–D**) Spatially averaged fluorescence of (**C**) Cal520HA-RBP (n=19) or (**D**) Cal520LA-RBP (n=30) as a function of time at ribbon proximal (light cyan), and distal (dark cyan) locations. Data points show the average intensity (± SEM) in each horizontal row of five pixels for 10 ms depolarizations at distinct ribbon locations. Fluorescence intensity at the onset of the 10 ms depolarizing stimulus (arrow) was normalized to the baseline fluorescence before stimulation and averaged over all pixels (i.e. over space and time). The amplitude differences between ribbon-proximal and ribbon-distal Ca$^{2+}$ signals were well-resolvable using the ribbon-bound Cal520HA-RBP indicator (**C**, light vs. dark cyan traces, paired t-test: $\Delta F/F_{rest}$ = 3.0±0.4 vs 1.9±0.3, respectively, p=0.001) and Cal520LA-RBP indicator (**D**, light vs. dark cyan traces, paired t-test: $\Delta F/F_{rest}$ = 5.5±0.9 vs 3.3±0.8, respectively, p=0.003). The current amplitudes were not significantly different between Cal520HA-RBP and Cal520LA-RBP readings (mean current amplitudes: Cal520HA-RBP=49.8±2.2 pA, Cal520LA-RBP=49.9±2.9 pA; p=0.99). (**C–D** inset) Temporal profile of events between 10–50 ms were expanded for better visualization. Scale bars: vertical, 1 ($\Delta F/F_{rest}$, **C** inset) or 2 ($\Delta F/F_{rest}$, **D** inset); horizontal, 10 ms. (**E–F**) Average fluorescence intensity of (**E**) proximal and (**F**) distal Ca$^{2+}$ signals obtained with Cal520LA-RBP (light cyan and dark cyan, respectively) and Cal520LA-free (gray). The amplitudes of ribbon-proximal Ca$^{2+}$ signals were higher when measured with Cal520LA-RBP than with Cal520LA-free (**E**, Cal520LA-RBP (light cyan) vs. Cal520LA-free (gray), unpaired t-test: $\Delta F/F_{rest}$ = 5.5±0.9, n=30 vs 3.1±0.4, n=21, p=0.04) but this was not the case for distal Ca$^{2+}$ signals (**F**, Cal520LA-RBP (light cyan) vs. Cal520LA-free (gray), unpaired t-test: $\Delta F/F_{rest}$ = 3.3±0.8, n=30 vs 1.9±0.2, n=21, p=0.15). (**E–F** inset) Events between 10–50 ms were expanded for better visualization. Scale bars: vertical, 2 ($\Delta F/F_{rest}$, **E** inset) and 1 ($\Delta F/F_{rest}$, **F** inset); horizontal, 10 ms.

the effective $K_{1/2}$ ($K_{eff}$) by measuring the Cal520HA-RBP/TAMRA-RBP fluorescence ratio in buffered Ca$^{2+}$ solutions and using the Grynkiewicz equation (*Grynkiewicz et al., 1985*). For Cal520HA-RBP, we found the $K_{eff}$ to be ~795 nM, which is higher than the value of 320 nM reported by the manufacturer for Cal520HA. The differences between the in-cell measurements and the manufacturer's values are likely to arise from differences in the cellular buffering capabilities, changes in dye properties due to

interactions with the molecules inside the cell (*Neef et al., 2018*; *Uto et al., 1991*; *Woehler et al., 2014*; *Shirakawa and Miyazaki, 2004*), and possible differences in the binding properties of the peptide-conjugated $Ca^{2+}$ indicators when bound to synaptic ribbons. Since the in-cell approach most closely matched the experimental conditions, we used this value ($K_{eff}$) for all further calculations.

We first measured the ribbon-proximal and ribbon-distal $Ca^{2+}$ concentrations using Cal520HA-RBP with 0.2 mM EGTA in the patch pipette, as it allowed us to resolve the gradient between the two signals (*Figure 1A*). Under these conditions, we found a maximum ribbon-proximal $Ca^{2+}$ concentration produced in response to a brief 10 ms pulse of 3.7 µM, and a maximum ribbon-distal $Ca^{2+}$ concentration of 0.7 µM. These *apparent* $Ca^{2+}$ concentration amplitudes are well below levels required to trigger exocytosis of the ultrafast releasable pool (UFRP) and readily releasable pool (RRP) (*Heidelberger et al., 1994*) and, as discussed earlier, likely represent the lower bounds of the $Ca^{2+}$ concentration at a single ribbon location due to local saturation of the high-affinity indicator and/or due to nanodomains that are smaller than the resolution attainable using light microscopy. It should also be noted that, unlike freely diffusing $Ca^{2+}$ indicators, RBP-conjugated indicators that slowly unbind from the ribbon are not readily replaced by free $Ca^{2+}$ indicators, rendering them even more prone to saturation. Nevertheless, our data demonstrate that ribbon-bound indicators may better report local $Ca^{2+}$ concentrations due to their localization, albeit still subject to the limitations of light microscopy.

To test the contribution of local saturation and to better estimate ribbon $Ca^{2+}$ concentrations, we repeated our measurements using the low-affinity $Ca^{2+}$ indicator Cal520LA conjugated to the RBP peptide. Because it was difficult to perform in-cell measurements to determine the $K_{eff}$ for Cal520LA-RBP given the large amounts of $Ca^{2+}$ required to calibrate the Cal 520LA indicator (see Methods and materials), we used the $K_{1/2}$ *value* provided by the manufacturer for free Cal520LA ($K_D$ 90 µM). However, we expect that in-cell measurements of $K_{eff,}$ are likely to be different due to the cellular buffering properties reported previously for $K_{eff,}$ measurements of Oregon Green BAPTA-5N in inner hair cells (*Neef et al., 2018*).

Bipolar cells release neurotransmitters primarily from ribbon active zones, although some release also occurs at non-ribbon sites (referred to as NR, *Figure 4A*; *Midorikawa et al., 2007*; *Zenisek, 2008*; *Chen et al., 2013*). To reveal the $Ca^{2+}$ signaling at and away from the ribbon, we performed whole-cell patch clamping and x-t line scans at ribbon (*Figure 4Ba*, R) and non-ribbon (*Figure 4Bb*, NR) sites perpendicular to the plasma membrane using TAMRA-RBP and Cal520LA-RBP. A brief (10 ms) depolarizing voltage-clamp pulse evoked rapid high $Ca^{2+}$ signals (*Figure 4Ba*, cyan raster plot) at ribbon locations (*Figure 4Ba*, magenta raster plot) but not at non-ribbon locations (*Figure 4Bb*, cyan raster plot). The amplitude of $Ca^{2+}$ signals elicited by brief depolarization and detected with Cal520LA-RBP at the ribbon-proximal site (*Figure 4C*, light cyan traces) were significantly higher than those at ribbon-distal (*Figure 4C*, dark cyan traces) and non-ribbon (*Figure 4C*, blue traces) sites. The $Ca^{2+}$ concentration gradient along the ribbon is summarized in *Figure 4D*. We found that the average $Ca^{2+}$ concentration at the proximal side of the ribbon (26.4±3.1 µM, n=26) was significantly different from that at the ribbon-distal (15.6±1.5 µM, n=26) and non-ribbon (10.4±0.4 µM, n=15) sites, and that $Ca^{2+}$ concentrations at ribbon-distal sites are higher than those at non-ribbon sites (*Figure 4D*). These measurements display large heterogeneity across distinct ribbons and distinct cells, with the coefficient of variation of about 60% for [$Ca^{2+}$] measurements at locations proximal to the ribbon, compared to a 10% CV for multiple depolarizations for the same ribbon. The $Ca^{2+}$ signals at distal sites also reached their peak amplitude earlier (*Figure 4C* inset, dark cyan arrowhead) than those at the non-ribbon sites (*Figure 4C* inset, blue arrowhead). These findings are consistent with non-ribbon vesicle release being governed by cytoplasmic residual $Ca^{2+}$ or $Ca^{2+}$ influx via clustered Cav channels at ribbon sites rather than non-ribbon active zones with clusters of Cav channels and are also in agreement with previous reports regarding the temporal delay and sensitivity of non-ribbon exocytosis to EGTA (*Zenisek, 2008*; *Mehta et al., 2014*).

## Sensitivity of microdomain $Ca^{2+}$ to exogenous buffers

Previous work has shown that the rate of vesicle replacement in RBCs is accelerated by elevated $Ca^{2+}$ levels at sites along the ribbon, and that while millimolar levels of EGTA have little effect on the fast exocytosis component near Cav channels, they selectively block sustained exocytosis, likely by preventing $Ca^{2+}$ from reaching the distal locations on the ribbon (*Mennerick and Matthews, 1996*; *Babai et al., 2010b*; *Gomis et al., 1999*; *Johnson et al., 2008*; *Singer and Diamond, 2006*). These

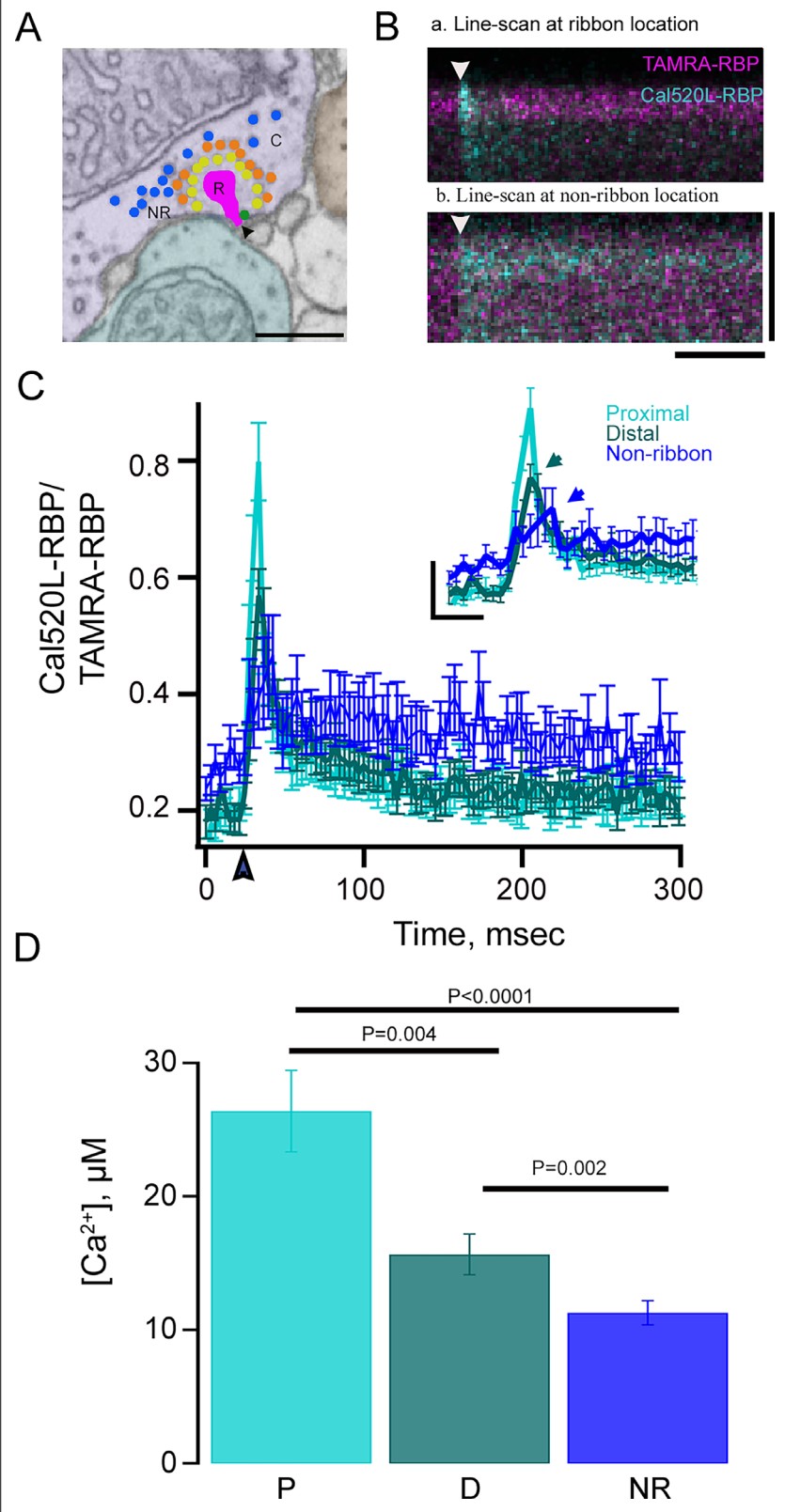

**Figure 4.** Ca²⁺ signals at synaptic ribbon at different distances from the plasma membrane. (**A**) Ultrastructure of a zebrafish rod bipolar cell (RBC) with kinetically distinct vesicle pools, as described in *Figure 1—figure supplement 1* ultrafast releasable pool (UFRP) (green vesicles), and readily releasable pool (RRP) (yellow vesicles) are primarily released via ribbon sites (**R**) at the cytomatrix of the active zone (arrowhead). Recycling pool (RP, orange vesicles)

*Figure 4 continued on next page*

*Figure 4 continued*

in the cytoplasm (**C**), likely to be released via non-ribbon (NR) sites. Scale bar: 500 nm. (**B**) Representative *x-t* plots show the fluorescence intensity of Cal520LA-RBP (cyan) and TAMRA-RBP (magenta) as a function of distance (vertical axis) and time (horizontal axis) at (**Ba**) ribbon sites and (**Bb**) non-ribbon (NR) locations. The darker region at the upper edge of each plot is the extracellular space and the arrowheads show the timing of depolarization. Scale bars: vertical, 1.6 µm and horizontal, 75 ms. (**C**) Spatially averaged Cal520LA-RBP as a function of time at ribbon proximal (light cyan), distal (dark cyan), and non-ribbon locations (blue) (n=26 (proximal ribbons), 26 (distal ribbons), and 15 (non-ribbon) from 5~8 different RBCs, respectively). (**C** inset) Temporal profile of events between 0–100 ms were expanded for better visualization. Scale bars: vertical, 0.2 ($\Delta F/F_{rest}$); horizontal, 20 ms. (**D**) $Ca^{2+}$ measurements along the ribbon axis using the nanophysiology approach demonstrate that the proximal $Ca^{2+}$ signals can go as high as 26.4±3.1 µM (light cyan, N=26) and distal as 15.6±1.5 µM (dark cyan, N=26), and non-ribbon 10.4±0.4 µM (NR, blue, N=15), respectively, in response to 10 ms stimuli. Error bars show standard errors. All conditions were significantly different from each other as assessed by paired t-test when comparing proximal vs. Distal and unpaired t-test when comparing non-ribbon to proximal or distal (proximal vs. distal *p*=0.004; proximal vs. non-ribbon *p*<0.0001; distal vs. non-ribbon *p*=0.002). The currents were not significantly different between conditions (Mean current: 0.2 mM EGTA Cal520LA-RBP proximal and distal = 51.8±3.2 pA, 0.2 mM EGTA Cal520LA-RBP non-ribbon=47.3±3.0 pA; *p*=0.35).

findings raise the possibility that $Ca^{2+}$ has two sites of action, one near the Cav channels that trigger vesicle release and one further away that replenishes the supply of releasable vesicles. *Burrone et al., 2002* proposed that endogenous $Ca^{2+}$ buffers regulate the size of the RRP by limiting the spatial spread of $Ca^{2+}$ ions and could suppress vesicle release at the periphery of the active zone in bipolar cells (*Burrone et al., 2002*). However, the $Ca^{2+}$ gradient at different locations along the ribbon controlling RRP release has never been examined. Thus, we used our higher resolution approach (*Figure 1*) to measure the $Ca^{2+}$ gradient along the synaptic ribbon under buffering conditions that differentially modulate vesicle release and resupply. To determine the spatiotemporal properties of $Ca^{2+}$ signals under these conditions, we performed rapid x-t line scans at a single ribbon location in the presence of ribbon-bound Cal520LA-RBP and under varying concentrations of exogenous buffer in the patch pipette solution, including EGTA at 0.2 (*Figure 5A*), 2 (*Figure 5B*), and 10 mM (*Figure 5C*) or BAPTA at 2 mM (*Figure 5D*). In these experiments, we chose Cal520LA-RBP to study the varying contributions of exogenous buffer, as Cal530HA-RBP measurements are less accurate. However, it should be noted that for briefer depolarizations and lower amplitude depolarizations, where the signal for the Cal520LA-RBP is small, the Cal520HA-RBP could be more useful. We found that the ratio between proximal and distal $Ca^{2+}$ signal amplitudes was similar in 0.2 and 2 mM EGTA (proximal-to-distal ratio ~1.7, *Supplementary file 1*) but was enhanced with 10 mM EGTA (proximal-to-distal ratio ~1.9, *Supplementary file 1*) and further enhanced with 2 mM BAPTA (proximal-to-distal ratio ~4.5, *Supplementary file 1*). These findings once again emphasize the reliable measurement of resolving ribbon-proximal vs. ribbon-distal $Ca^{2+}$ signals using our nano-physiology approach. Our experimental findings of the increase in the proximal-to-distal $Ca^{2+}$ concentration ratio with increasing EGTA concentration are consistent with our simulation results (Figure 7), although the corresponding ratios are greater in the simulation than in the experiment due to the large size of the microscope's point-spread function.

Though BAPTA significantly abolished the spread of $Ca^{2+}$ signals, it is impressive that a substantial amount of ribbon-proximal $Ca^{2+}$ concentration measured with ribbon-bound indicators was still present with 2 mM BAPTA. One reason for this observation could be that ribbon-bound $Ca^{2+}$ indicators are likely to measure $Ca^{2+}$ signals very close to Cav channels without diffusing away, which makes it impossible for BAPTA to intercept $Ca^{2+}$ ions. If so, similar experiments conducted with free $Ca^{2+}$ indicators should report a significantly lower proximal $Ca^{2+}$ signal since the measurement by a diffusible dye would effectively spread the signal over a larger volume. Indeed, this is what we found. As shown in *Figure 5E*, the apparent proximal $Ca^{2+}$ signal in response to a 10 ms brief pulse measured with Cal520LA-free in the presence of 2 mM BAPTA was 1.4±0.2 (n=20), 2.6-fold lower than proximal $Ca^{2+}$ signals measured with Cal520LA-RBP (3.6±1, n=20). The latter value is closer to the true measure of the $Ca^{2+}$ concentration in the vicinity of the ribbon base. These findings emphasize that the reliable measurement of ribbon-proximal $Ca^{2+}$ signals in the vicinity of Cav channels greatly benefits from the increased spatial resolution of our nano-physiology approach. We also measured the spatiotemporal properties of local $Ca^{2+}$ signals using Cal520HA-RBP, as we have done above for Cal520LA-RBP

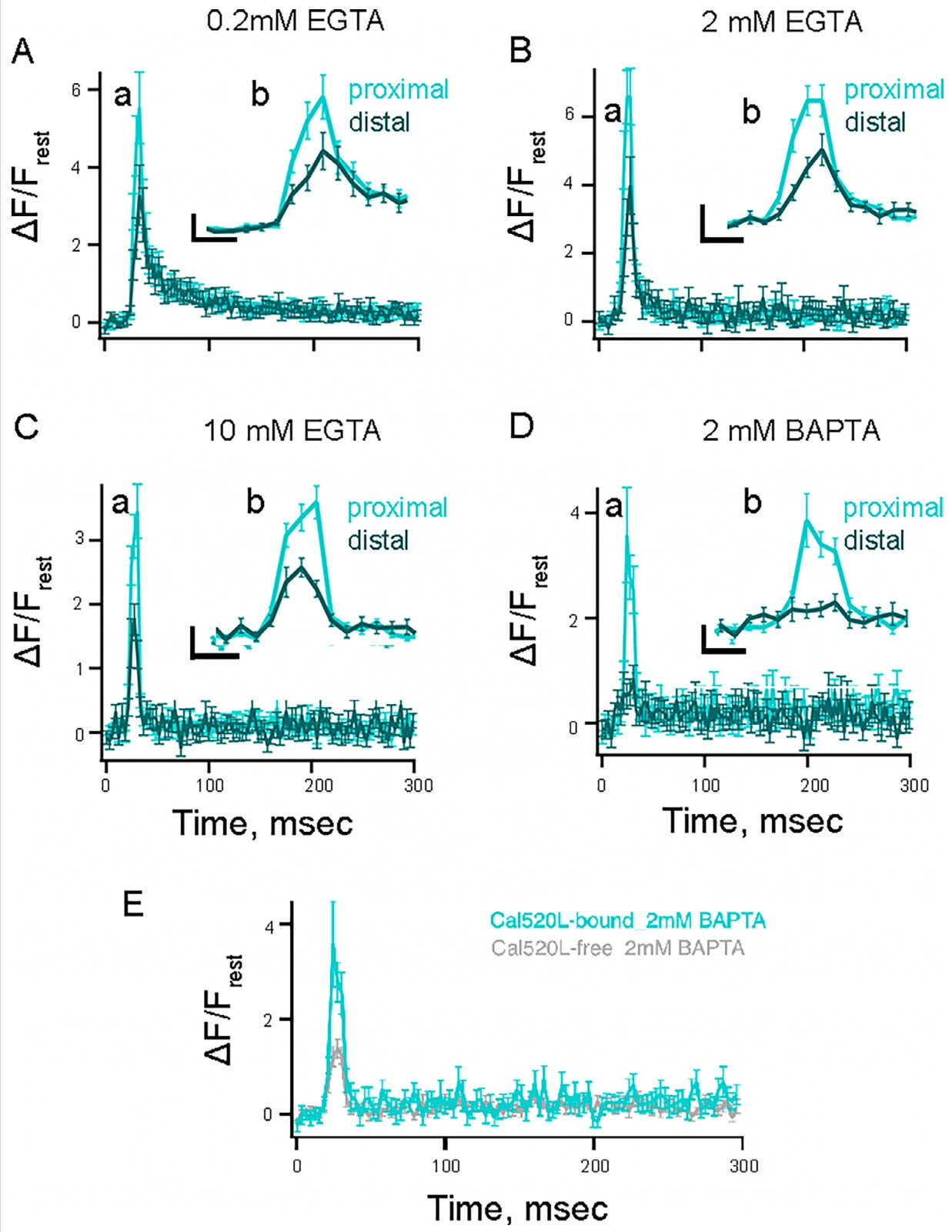

**Figure 5.** Effect of exogenous $Ca^{2+}$ buffers on spatiotemporal properties of $Ca^{2+}$ microdomains in rod bipolar cell (RBC) terminal recorded with low-affinity ribbon-bound dye. (**A–D**) Average temporal fluorescence intensity (normalized to $\Delta F/F_{rest}$) of proximal (light cyan) and distal (dark cyan) $Ca^{2+}$ signals with Cal520LA-RBP as a function of time at distinct ribbon locations with pipette solution containing (**A**) 0.2 mM EGTA, (**B**) 2 mM EGTA, (**C**) 10 mM EGTA, or (**D**) 2 mM BAPTA. Proximal measurements were significantly higher than distal measurements in all conditions as assessed using

*Figure 5 continued on next page*

*Figure 5 continued*

paired t-tests (0.2 mM EGTA: p=0.0027, n=30; 2 mM EGTA: *p*=0.034, n=21; 10 mM EGTA: *p*<0.001, n=43; 2 mM BAPTA: p=0.0073, n=20). The currents between conditions were not significantly different from each other (mean current amplitudes in 0.2 mM EGTA: 50.6±3.0 pA, 2 mM EGTA: 49.7±3.1 pA, 10 mM EGTA: 47.3±3.1 pA, 2 mM BAPTA: 56.1±2.6 pA; 0.2 mM EGTA vs 2 mM EGTA: *p*=0.84, 0.2 mM EGTA vs 10 mM EGTA: *p*=0.46, 0.2 mM EGTA vs 2 mM BAPTA: *p*=0.19). Inset. The temporal profiles of events between 0–50 ms were expanded for better visualization. Scale bars: vertical, 2 ($\Delta F/F_{rest}$; panels **A** and **B**) and 1 ($\Delta F/F_{rest}$; panels **C** and **D**); horizontal, 10 ms. (**E**) Average temporal fluorescence intensity (normalized to $\Delta F/F_{rest}$) of proximal $Ca^{2+}$ signals measured with Cal520LA-RBP (cyan) and Cal520LA-free (gray) as a function of time with pipette solution containing 2 mM BAPTA. The corresponding maximum values of trial-averaged $\Delta F/F_{rest}$ was significantly higher with 2 mM Cal520LA-bound BAPTA than 2 mM Cal520LA-free BAPTA, with mean ± SEM of 3.6±0.9 and 1.4±0.2, respectively (unpaired t-test, p=0.028; Cal520LA-free: n=20; Cal520LA-bound: n=20). The currents between conditions were not significantly different from each other (mean current amplitudes in 2 mM BAPTA Cal520LA-RBP: 56.1±2.6 pA; 2 mM BAPTA Cal520LA-Free: 53.3±2.2 pA; *p*=0.42).

The online version of this article includes the following figure supplement(s) for figure 5:

**Figure supplement 1.** Spatiotemporal properties of $Ca^{2+}$ microdomains along the synaptic ribbon in the rod bipolar cell (RBC) terminal.

(*Figure 5—figure supplement 1*). We found that the ratio between proximal and distal $Ca^{2+}$ signals was similar in 0.2 and 2 mM EGTA (proximal-to-distal ratio ~1.5, *Supplementary file 2*) but was enhanced with 10 mM EGTA (proximal-to-distal ratio ~2, *Supplementary file 2*) and further enhanced with 2 mM BAPTA (proximal-to-distal ratio ~2.8, *Supplementary file 2*). Cal520 acts as a buffer that binds $Ca^{2+}$ and carries it away by diffusion. Thus, the higher affinity indicator Cal520HA will bind and shuttle away more $Ca^{2+}$ than Cal520LA. If the Cal520HA indicator is saturated near the Cav channels, it may be underreporting the fast $Ca^{2+}$ transients that occur in those locations. However, Cal520LA and Cal520HA show similar increase in the ribbon proximal-to-distal ratios with increasing concentrations of exogenous buffer.

## Computational models of $Ca^{2+}$ signals along the axis of the RBC synaptic ribbon

Since we could identify the position of the ribbon and we used RBP-fused $Ca^{2+}$ indicator-RBP, we were able to measure and distinguish ribbon-proximal vs. ribbon-distal $Ca^{2+}$ signals that drive the release of UFRP and RRP. However, several factors, including finite spatiotemporal resolution of optical imaging, spatial diffusion, dye saturation, and dye binding kinetics, may limit our ability to achieve optimal resolution. These limitations, however, can be overcome by the use of quantitative models. Thus, we developed a model based on our data with published information on $Ca^{2+}$ diffusion and buffering to estimate more accurately the $[Ca^{2+}]_i$ gradient along the ribbon at distinct distances from the plasma membrane.

Combining whole-terminal $Ca^{2+}$ current measurements and our estimate for the number of synaptic ribbons per terminal allowed us to infer the magnitude of $Ca^{2+}$ current per single ribbon, which we used in our model to determine the spatiotemporal $[Ca^{2+}]$ dynamics near the ribbon. Figure 7 shows the results of the simulation of $[Ca^{2+}]$ at various distances from the ribbon during and after a depolarizing pulse of 10 ms duration in the presence of different concentrations of exogenous and endogenous buffers, replicating the conditions used in our experiments. The geometry of the simulation domain box is shown in *Figure 6*; it represents the fraction of total synaptic terminal volume per single ribbon (see Methods, *Supplementary file 3* and *Figure 6—figure supplement 1*). *Figure 7* shows the time-dependence of $[Ca^{2+}]$ during and after the depolarization pulse at 5 specific locations both near and distant from the ribbon (left-hand panels in each subplot). These five spatial locations are marked by circles of different colors in the right-hand panel of each subplot, which shows $[Ca^{2+}]$ at the end of the $Ca^{2+}$ current pulse in pseudo-color in the entire planar cross-section of the simulation domain cutting through the middle of the ribbon, as shown in *Figure 6*. We assumed a highly simplified Cav channel arrangement into four clusters forming a square with a side length of 80 nm. The closest of the five spatial locations (red curves and circles in *Figure 7*) is X=20 nm away from the base of the ribbon center-line, Z=10 nm above the ribbon, about 45 nm away from the closest Cav channel cluster (see *Figure 6*). We assumed that this location was within the $Ca^{2+}$ microdomain that triggered the release of the UFRP. Vesicles were not included in the simulation since the total exclusion volume attributed to them represented only a small fraction of the total inter-terminal volume and, therefore, did not significantly impact $[Ca^{2+}]$ at the qualitative resolution level we are interested in. A couple of simulations with vesicles included were performed to confirm this statement, requiring much finer

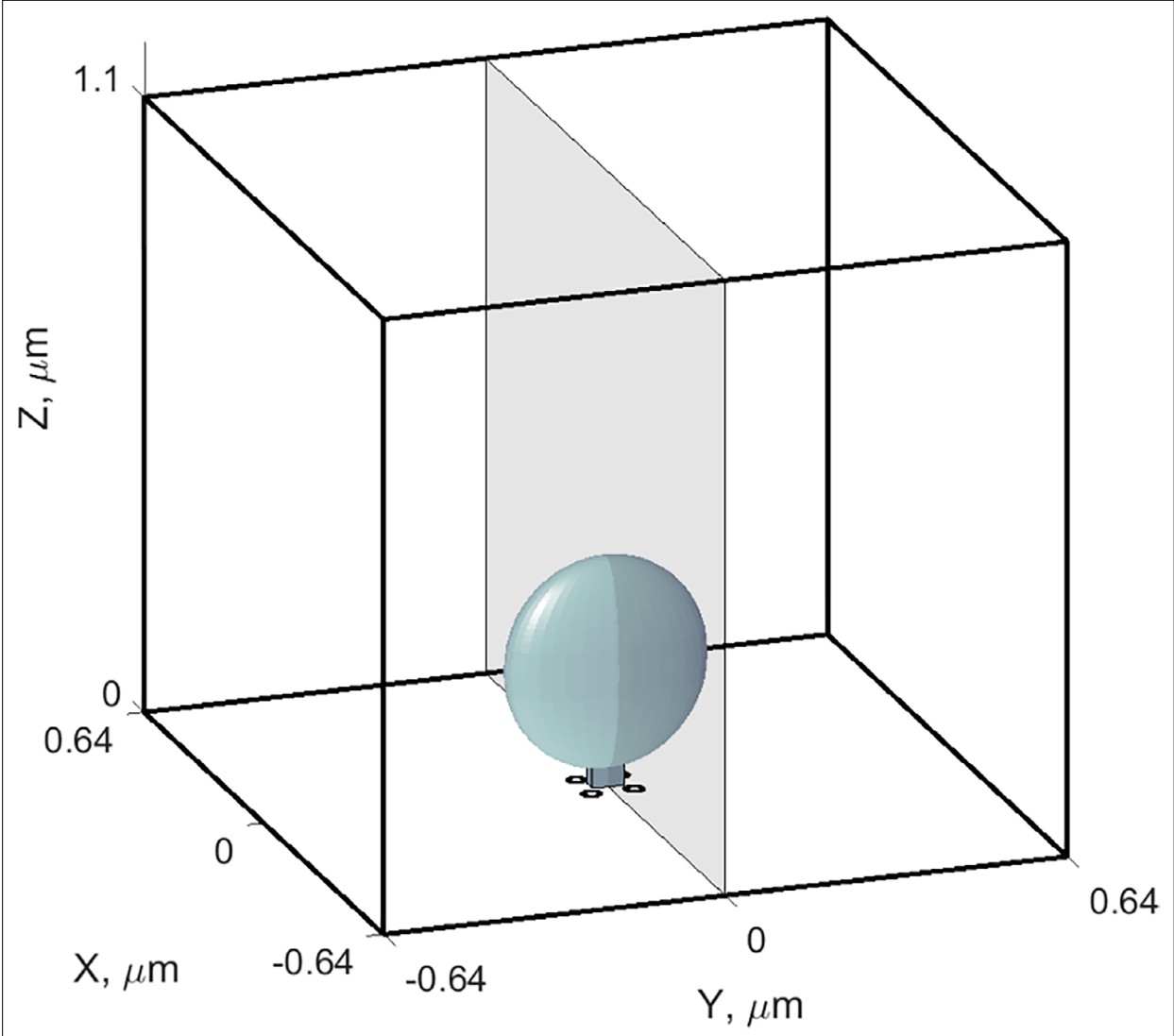

**Figure 6.** Geometry of the computational model of intra-terminal Ca²⁺ dynamics. Simulation domain is a box with dimensions (1.28×1.28 ×1.1) µm³, approximating the fraction of synaptic terminal volume per single ribbon. Ca²⁺ ions enter near the base of the ellipsoidal ribbon at four locations marked by black disks representing Ca²⁺ channels or their clusters. The semi-transparent gray coordinate plane Y=0 corresponds to the section used for the pseudo-color 2D [Ca²⁺] plots in *Figure 7*. Ca²⁺ is extruded on all surfaces of this domain, simulating combined clearance by pumps and exchangers on the plasmalemmal as well as internal endoplasmic reticulum and mitochondrial membranes.

The online version of this article includes the following figure supplement(s) for figure 6:

**Figure supplement 1.** Simulated effect of the endogenous mobile buffer of different concentrations on [Ca²⁺] dynamics in response to a 10 ms pulse.

spatial resolution and longer computational time. We note also that [Ca²⁺] for short pulse durations considered here is relatively insensitive to our assumptions on the membrane Ca²⁺ extrusion mechanisms, listed in *Supplementary file 3*.

*Figure 7A1–B2* reveals that [Ca²⁺] could rise above 50 µM within the microdomain near the base of the ribbon in the presence of 0.2 mM up to 10 mM EGTA. Given the size of the microscope point-spread function, this is in qualitative agreement with the 26 µM estimate of ribbon-proximal [Ca²⁺] that we recorded using the ribeye-bound low-affinity indicator. In addition, this concentration level is reached very soon after the channel opening event due to the rapid formation of the Ca²⁺ micro-domain (*Eisner et al., 2023*; *Neher, 1998*), and therefore, this estimate is expected to hold for shorter pulses as well. However, [Ca²⁺] decayed rapidly with distance, with the rate of decay significantly increasing when EGTA concentration was increased to 10 mM (*Figure 7B1, B2*). Note that a sevenfold change in the capacity of the immobile endogenous Ca²⁺ buffer (*Figure 7A1 vs 2*) had a relatively

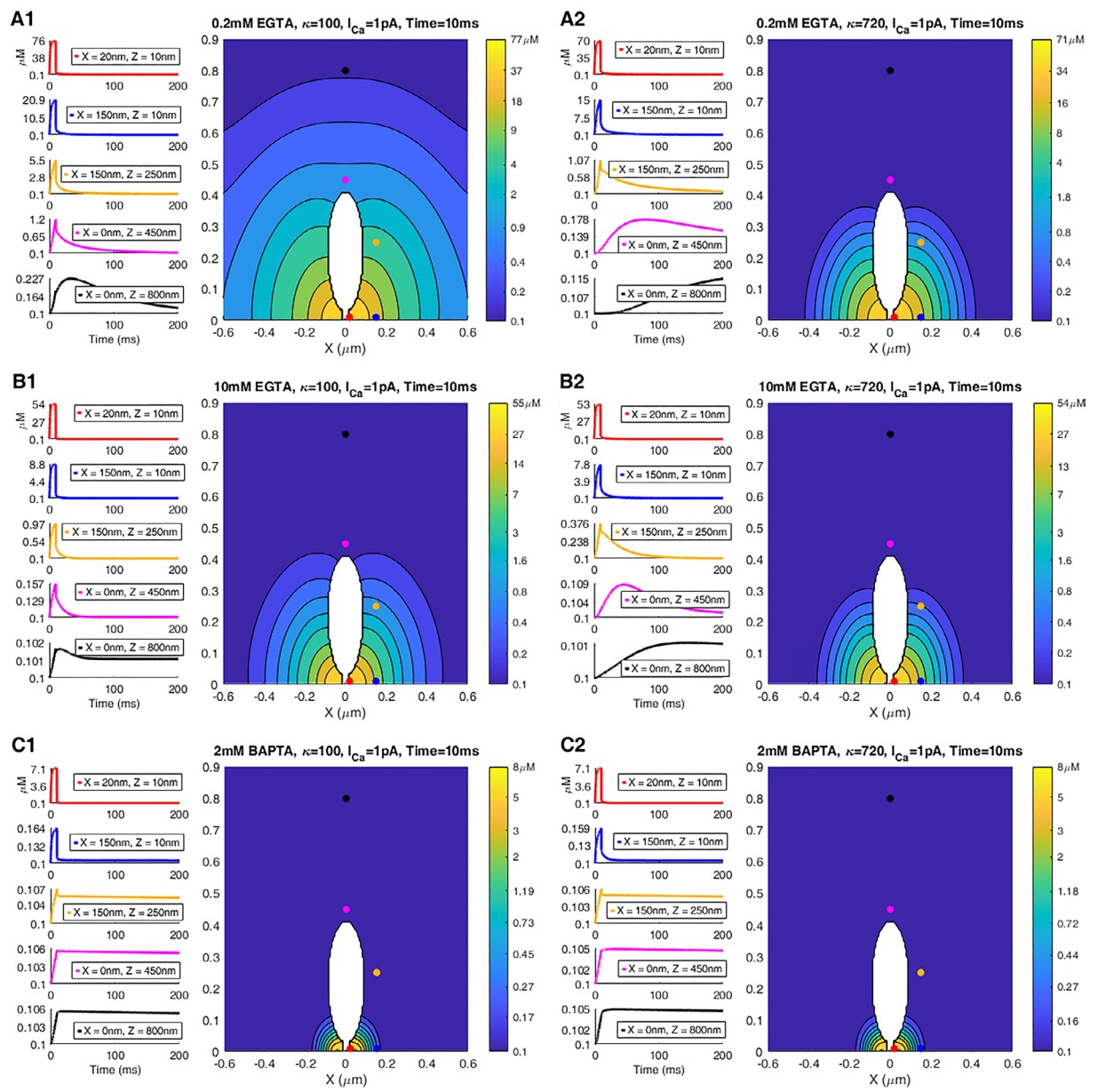

**Figure 7.** Simulation of the effect of an endogenous immobile buffer of different concentrations on [Ca²⁺] dynamics in response to a 10 ms pulse. (**A1–C2**) In each sub-plot, the left-hand panels show the [Ca²⁺] time course in response to a 10 ms constant current pulse (total current of 1 pA) at five select locations marked by colored circles in the right panel. The right-hand panels show a pseudo-color plot of [Ca²⁺] in a 2D section of the 3D simulation volume in *Figure 6*, at a fixed point in time corresponding to the end of the current pulse. Concentration values for each level curve are indicated in the color bar. Endogenous buffer is immobile, with a concentration of 200 μM in panels **A1**-**C1** (resting buffering capacity 100 μM), vs. 1.44 mM in panels **A2**-**C2** (resting buffering capacity 720). Exogenous buffer concentrations were 0.2 mM EGTA (panels **A1**, **A2**), 10 mM EGTA (panels **B1**, **B2**), 2 mM BAPTA (panels **C1**, **C2**).

modest effect on the [Ca²⁺] level up to about 150 nm distance from the ribbon base. This effect of increasing endogenous buffer concentration was further reduced in 10 mM EGTA (*Figure 7B1 vs 2*), due to strong competition of endogenous buffer with large concentrations of EGTA. This agrees with the expectation that immobile buffers primarily slow down Ca²⁺ signals but do not affect the rapidly forming quasi-steady-state Ca²⁺ microdomains in the immediate vicinity of the channel (*Eisner et al., 2023*; *Neher, 1998*; *Matthews and Dietrich, 2015*). Finally, *Figure 7C1-C2* shows that 2 mM BAPTA had a much greater effect on localizing the Ca²⁺ signal to the immediate vicinity of the ribbon base.

Since a native unpatched cell may well contain mobile rather than immobile buffers, it was interesting to examine the effect of increasing the concentration of mobile endogenous buffer on $Ca^{2+}$ dynamics in the ribbon vicinity, as compared to the corresponding effect of immobile buffer. *Figure 6—figure supplement 1B–C* shows that a sevenfold increase in mobile buffer concentration reduces the microdomain $Ca^{2+}$ by a factor of 2 and greatly reduces the size of the microdomain. This contrasts with the effect of increasing the concentration of the immobile buffer, which has a much more subtle effect (*Figure 7A1 and A2*). We note that 0.2 mM EGTA was absent in the simulation with mobile buffer shown in *Figure 6—figure supplement 1B–C*; in general, the effect of modest concentrations of EGTA is expected to be negligible due to its slow $Ca^{2+}$ binding speed compared to the endogenous buffer.

## Variability of local $Ca^{2+}$ signals across the RBC synaptic ribbons

Bipolar cells release neurotransmitters primarily from ribbon active zones (*Matthews and Fuchs, 2010*; *Oesch and Diamond, 2011*; *Coggins and Zenisek, 2009*; *Mennerick and Matthews, 1996*; *Singer and Diamond, 2003*; *Snellman et al., 2009*; *Datta et al., 2017*; *Euler et al., 2014*; *Neves and Lagnado, 1999*; *von Gersdorff and Matthews, 1997*; *Zhou et al., 2006*). However, the factors that shape the synaptic ribbon microdomains at retinal ribbon synapses have not been examined. We wondered whether all of the 30–50 synaptic ribbons at an RBC terminal release glutamate in a similar fashion or whether there is some heterogeneity in glutamate release from different ribbon active zones. In particular, we asked what underlying mechanisms could differentiate the $Ca^{2+}$ signals between ribbons of the same RBC terminal and across different RBCs. We first found evidence for such variability in local $Ca^{2+}$ signals at single-ribbon locations of different cells using the freely diffusible high-affinity indicator Cal520HA, observing high variability in the $Ca^{2+}$ transient amplitudes, even for RBCs with similar depolarization-evoked $Ca^{2+}$ current amplitudes (*Figure 8—figure supplement 1A vs. 1B*). However, we did not observe such variability when we obtained multiple line scans across the same ribbon (*Figure 8—figure supplement 1A vs. 1B*, gray traces). We wondered whether the observed variability between cells could be due to distinct subtypes of zebrafish RBCs (*Hellevik et al., 2024*), which might have different subtypes or numbers of Cav channels near the ribbon. Recent work in zebrafish retina identified two distinct RBC subtypes: RBC1 and RBC2 (*Hellevik et al., 2024*). The wiring pattern of amacrine cells postsynaptic to RBC1 closely resembles the circuitry of mammalian RBCs, whereas RBC2 forms distinct pathways. Furthermore, RBC1 specifically labels for the known marker of mammalian RBCs, PKC-α (*Hellevik et al., 2024*). Due to these similarities, RBC1 in zebrafish is classified as analogous to mammalian RBCs. Moreover, RBC1s have expected morphological characteristics, in particular, the shape and size of soma and the presence of a single synaptic terminal (*Hellevik et al., 2023*). Immunolabeling of isolated RBC preparation used in these experiments showed primarily intact RBCs, which are PKC-α-specific RBC1 (*Figure 1—figure supplement 2B*). For simplicity, and because this study focuses exclusively on RBC1, we will refer to RBC1 as 'RBC' throughout this work. Thus, we attribute the variability in the RBC $Ca^{2+}$ transients to the variability between the $Ca^{2+}$ microdomains within the same cell type.

To identify mechanisms that contribute to the heterogeneity in the local $Ca^{2+}$ signals we have reported in this study, we began by asking whether differences in local $Ca^{2+}$ buffering could account for this heterogeneity. To examine the role of $Ca^{2+}$ buffering in the variability of local $Ca^{2+}$ signals across different RBC and different ribbons (*Figure 8A and D,* respectively), we compared proximal (*Figure 8B and E*) and distal (*Figure 8C and F*) $Ca^{2+}$ signals in 10 mM EGTA by averaging the $Ca^{2+}$ recordings from all the ribbons of a given RBC (*Figure 8—figure supplements 2 and 3* and *Figure 8—source data 1 and 2*). Here, we first compare proximal (panel B) and distal (panel C) calcium signals across several RBCs, labeled RBC-a through RBC-d. Each RBC contains multiple ribbons, and for each cell, we present the average calcium signals from multiple ribbons using box plots in panels B and C. In these box plots, the horizontal lines represent the average calcium signal for each cell, while the size of the error bars reflects the variability in proximal and distal calcium signals among the ribbons within that RBC. For example, RBC-a had five identifiable ribbons. When the $Ca^{2+}$ signals were restricted to the ribbon sites with 10 mM EGTA in the pipette solution (*Supplementary files 1 and 2*), the amplitude of the ribbon-proximal and ribbon-distal $Ca^{2+}$ signals averaged over all ribbons of a given cell exhibited significant variation across different RBCs (*Figure 8B and C*, *Figure 8—figure supplement 2* and *Figure 8—source data 1*). In *Figure 8D–F*, we use RBC-a to illustrate the variability in $Ca^{2+}$

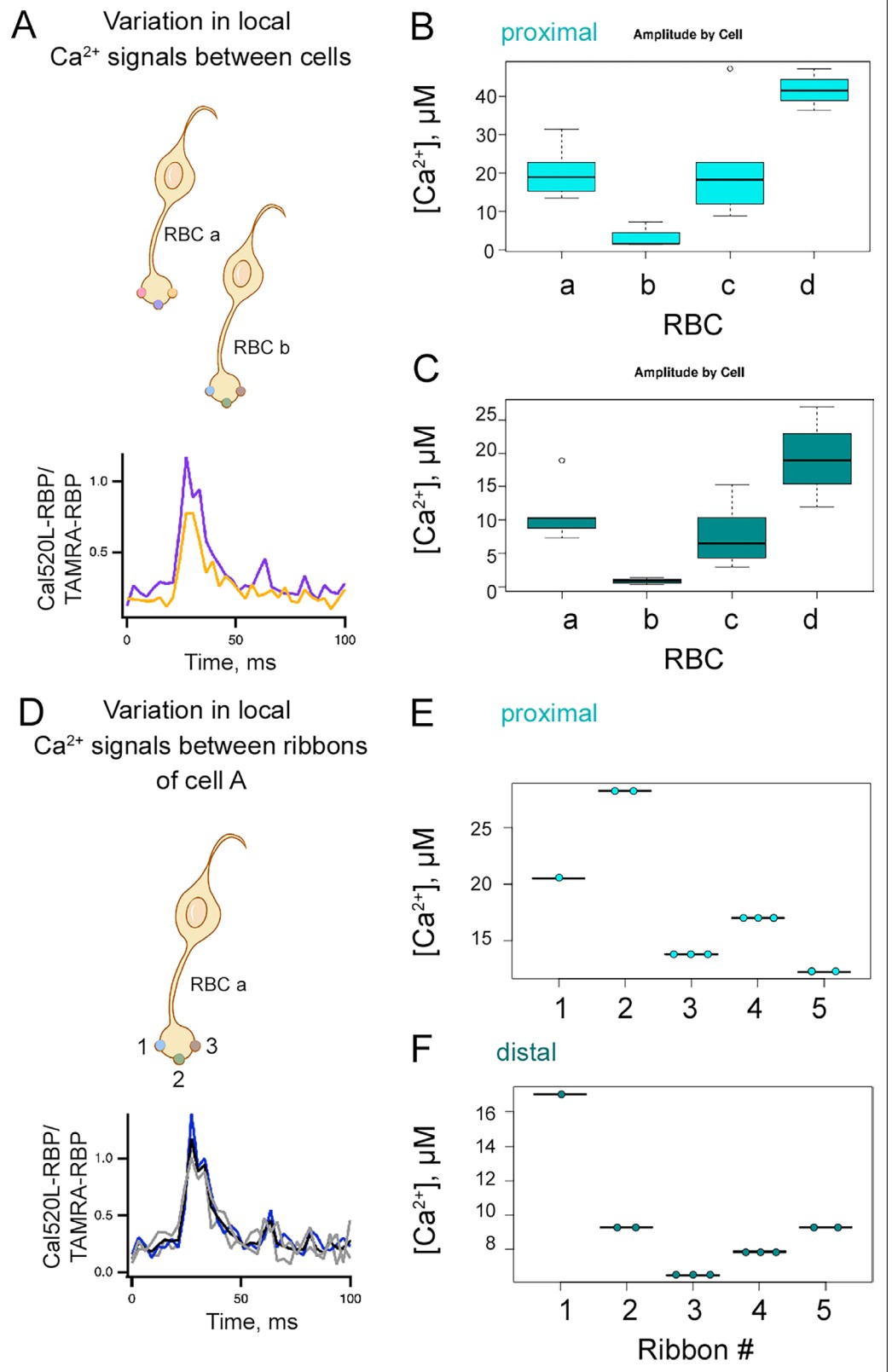

**Figure 8.** Heterogeneity in the spatiotemporal properties of $Ca^{2+}$ microdomains in rod bipolar cell (RBC) terminal. (**A**, Top panel) Cartoon of two representative RBCs (cell A and cell B), each containing differently colored ribbons. Bottom panel. Ribbon-to-ribbon variability was measured by recording local $Ca^{2+}$ signals near different ribbons (yellow and purple traces) for each RBC. If an RBC had multiple readings for a single ribbon, averages were

*Figure 8 continued on next page*

*Figure 8 continued*

obtained for comparisons as described in *Figure 8—figure supplement 2*. Bottom panel inset: sample $Ca^{2+}$ currents for the cells from which the $Ca^{2+}$ signal sample traces mentioned above were obtained (purple and yellow traces). Currents were similar across the different cells. Vertical scale = 80 pA, horizontal scale = 5 ms. (**B–C**) Variability of $Ca^{2+}$ signals in different ribbons across different cells in the presence of 10 mM EGTA, recorded using Cal520LA-RBP. Proximal $Ca^{2+}$ amplitude values were significantly different between cells **A** and **B** ($p$=0.015), cells **A** and **D** ($p$=0.016), and cells **B** and **D** ($p$=0.004) but similar between A and C (n=4 cells, 4 fish), as assessed by Welch's ANOVA with the Games-Howell post-hoc test. (**C**) In distal locations, ribbon amplitude values were significantly different between cells **A** and **B** ($p$=0.023) and cells **B** and **C** ($p$=0.049), but similar across all other cell comparisons (n=4 cells, 4 fish), as assessed by Welch's ANOVA with the Games-Howell post-hoc test. The currents were not significantly different across the different cells, as assessed by unpaired t-tests (mean currents in RBC a: 47.7±2.9 pA, RBC b: 43.7±4.0 pA, RBC c: 43.4±2.0 pA, RBC d: 40.9±2.4 pA; RBC a vs. RBC b: $p$=0.45, RBC a vs. RBC c: $p$=0.22, RBC a vs. RBC d: $p$=0.14, RBC b vs. RBC c: $p$=0.93, RBC b vs. RBC d: $p$=0.54, RBC c vs. RBC d: $p$=0.52).(**D**, Top panel) Illustration of a RBC containing three ribbons (numbered 1–3). Bottom panel. $Ca^{2+}$ signal measurements from three distinct ribbons (black, gray, and blue traces) were compared to determine the ribbon-to-ribbon variability within each RBC, as described in *Figure 8—figure supplement 3*. Bottom panel inset: sample $Ca^{2+}$ currents for the cells from which the $Ca^{2+}$ signal sample traces mentioned above were obtained (black, gray, and blue traces). Currents were similar across the different cells. Vertical scale = 80 pA, horizontal scale = 5 ms. (**E–F**) Box plot illustrating $[Ca^{2+}]$ across various ribbons of an individual RBC, which is shown as RBC a. Ribbon variability within individual cells was measured with 10 mM EGTA using Cal520LA-RBP at (**E**) proximal and (**F**) distal locations. (**E**) Proximal $Ca^{2+}$ amplitude values were significantly different among all ribbons (paired t-test, $p$<0.001) (n=5 ribbons, 1 RBC, 1 fish). (**F**) Distal $Ca^{2+}$ amplitudes were significantly different among all ribbon comparisons (paired t-test, $p$<0.001) except for ribbons 2 and 5 (n=5 ribbons, 1 RBC, 1 fish). Similar analyses were conducted in two more cells and found similar observations (data not shown). The currents were similar across all ribbons since these were readings from the same cell. Given that some ribbons only have one reading, it is not possible to conduct a paired t-test to statistically compare them; however, the average current ± standard error for the cell shown was 47.7±2.9 pA.

The online version of this article includes the following source data and figure supplement(s) for figure 8:

**Source data 1.** Data presentation for ribbon variability between cells.

**Source data 2.** Data presentation for ribbon variability within individual cells.

**Figure supplement 1.** Variability in $Ca^{2+}$ transients in response to brief stimuli.

**Figure supplement 2.** Illustration to demonstrate ribbon variability between cells.

**Figure supplement 3.** Illustration to demonstrate ribbon variability within individual cells.

signals across individual ribbons. Specifically, we distinguished proximal and distal $Ca^{2+}$ signals from five ribbons (ribbons 1–5) within RBC-a. We next compared variability in $Ca^{2+}$ signals in the presence of 10 mM EGTA at individual ribbons within the same cell (*Figure 8D*, *Figure 8—figure supplement 3* and *Figure 8—source data 2*) at proximal (*Figure 8E*) and distal (*Figure 8F*) locations. The box plots in *Figure 8E and F* display the average $Ca^{2+}$ signal (horizontal lines) for each ribbon, based on multiple recordings. For the cell described in *Figure 8D-F*, the proximal $Ca^{2+}$ signals were significantly different across all ribbons examined, and there were considerable differences in distal $Ca^{2+}$ signals between ribbons, with the exception of ribbons numbered 2 and 5. Importantly, the lack of or minimal error bars for repeated measurements at the same ribbon indicates that the proximal and distal calcium signals are consistent within a ribbon. These findings emphasize that the observed variability among ribbons and among cells reflects true biological heterogeneity in local calcium domains, rather than experimental noise. The heterogeneity in proximal and distal $Ca^{2+}$ signals at distinct ribbons within the same cell may result from different underlying mechanisms, for example, heterogeneity in Cav expression or subtype and $Ca^{2+}$ handling mechanisms. These findings suggest that exogenous $Ca^{2+}$ buffering has a negligible effect on experimentally observed heterogeneity and variability of the proximal $Ca^{2+}$ signals and that local $Ca^{2+}$ signals at RBC ribbons are dominated by $Ca^{2+}$ in regions close to the ribbon base where Cav channels are located.

## The ultrastructure of the zebrafish RBC terminal reveals diversity in the size of the synaptic ribbons across the terminal

In hair cells, $Ca^{2+}$ microdomain signaling varies with ribbon size, reflecting larger patches of Cav channels aligned with larger ribbons (*Frank et al., 2009*). However, the number, size, and shape of ribbon

active zones per terminal have not been established in our experimental system, the synaptic terminals of zebrafish RBCs. To provide quantitative measurement of ribbon microdomains, we examined the ultrastructure of zebrafish RBC synaptic ribbons by serial block face scanning electron microscopy (SBF-SEM) and reconstructed three RBCs from serial section electron micrograph (*Figure 9A*). We analyzed the bipolar cell terminals that are closest to the ganglion cell layer with a morphology similar to mammalian RBCs and zebrafish RBC1s (*Hellevik et al., 2024*). Our reconstruction of three RBCs revealed 30–41 ribbons within RBC terminals (*Figure 9A*), similar to what was previously reported in goldfish giant ON-type mixed RBCs (range 45–60, *von Gersdorff et al., 1996*).

We examined the distribution of these 30–41 synaptic ribbons within the zebrafish RBC terminals. The distribution of RBC ribbons as estimated by the distance between ribbon sites (*Figure 9B* and *Video 1*; see *Materials and methods*) revealed a wide distribution in synaptic ribbon distance to its nearest neighbor across the three reconstructed RBCs (means of 2.48±0.12 µm, 1.84±0.06 µm, and 1.98±0.08 µm; *Figure 9C*). The distance between RBC synaptic ribbons ranged from 141 nm to 6 µm, with a mean of 2±0.05 µm (*Figure 9D*; across all three reconstructed RBCs). Of the 510 comparisons of zebrafish RBC synaptic ribbons, only four ribbon pairs (0.8%) were separated by distances smaller than our confocal microscope resolution limit of 270 nm.

To reveal whether heterogeneity in synaptic $Ca^{2+}$ signals correlates with active zone size, we compared the shape and size of the ribbon active zones across the three reconstructed RBCs. Our results show significant variations in the shape and size of zebrafish RBC synaptic ribbons in RBCs and associated active zones (*Figure 10*). On average, individual ribbons spanned 2–5 consecutive sections, with some located within the axons or closer to the terminal. The SBF-SEM images and 3D projections of ribbon structures (colored in magenta) revealed considerable variability in shape and size across all dimensions, as illustrated in *Figure 10A–C*. The number of Cav channels per active zone in hair cells from chicken, frog, and turtle varies with the size of the synaptic ribbon (*Martinez-Dunst et al., 1997*). We thus measured the area of the ribbon facing the plasma membrane where the Cav channels are known to be located to estimate the number of Cav channels per active zone and to determine whether variation in active zone size might plausibly contribute to the heterogeneity in $Ca^{2+}$ signaling. The EM images and 3D projection of the plasma membrane (yellow) and the area of the ribbon facing the plasma membrane (cyan), representing the active zone, are shown in *Figure 10A–C*. We observed large variability in the area of the ribbon facing the plasma membrane or active zone within the RBCs, with substantial variability in their average distribution across the three RBCs (*Figure 10C*, cyan, *Figure 10E*, *Video 2*, and *Figure 10—figure supplement 1*), suggesting that the number of Cav channels per RBC active zone could plausibly be heterogeneous.

## The size of the active zone and maximal $Ca^{2+}$ influx correlate with the size of the synaptic ribbon

To reveal whether larger ribbons display stronger maximal $Ca^{2+}$ influx, we measured $Ca^{2+}$ signals in response to a series of 200 ms depolarizations. We first imaged TAMRA-RBP and Cal520LA-RBP in sequential scans to obtain ribbon location and resting $Ca^{2+}$ levels. To maximize the capture of $Ca^{2+}$ signals during brief stimuli, we image the Cal520LA-RBP channel, followed by TAMRA-RBP and Cal520LA-RBP sequential scans to confirm the ribbon locations. We found that the maximum amplitude of depolarization-evoked $Ca^{2+}$ signals increased with an increase in ribeye fluorescence (*Figure 11A & B* and *Video 3*; $r$=0.51, N=122 ribbons, $p$<0.001), consistent with findings reported in cochlear inner hair cells (*Frank et al., 2009*; *Ohn et al., 2016*). Since ribeye fluorescence correlates with the number of ribeye molecules per ribbon and, therefore, ribbon size (*Zenisek et al., 2004*; *Frank et al., 2009*; *Ohn et al., 2016*; *Wong et al., 2014*), our findings suggest that larger ribbons display stronger $Ca^{2+}$ signals as active zone size scales with ribbon size (*Ohn et al., 2016*). Given the diverse shapes of RBC synaptic ribbons (*Figure 10A & B*, magenta), we measured the longest length and width of ribbons from EM images and compared this with the active zone size. We found that RBC ribbon length and width have a moderate positive correlation (*Figure 11C*, $r$=0.47, N=102 ribbons, $p$<0.001), and both dimensions have a moderate positive correlation with active zone size, albeit the ribbon width (*Figure 11D*, $r$=0.52, N=102 ribbons, $p$<0.001) has a stronger correlation with active zone size than the ribbon length (*Figure 11D*, $r$=0.32, N=102 ribbons, $p$<0.001). These results suggest that heterogeneity in synaptic $Ca^{2+}$ signals correlates with ribbon dimensions and that active zone and the number of Cav channels (*Frank et al., 2009*; *Ohn et al., 2016*) scale with ribbon size.

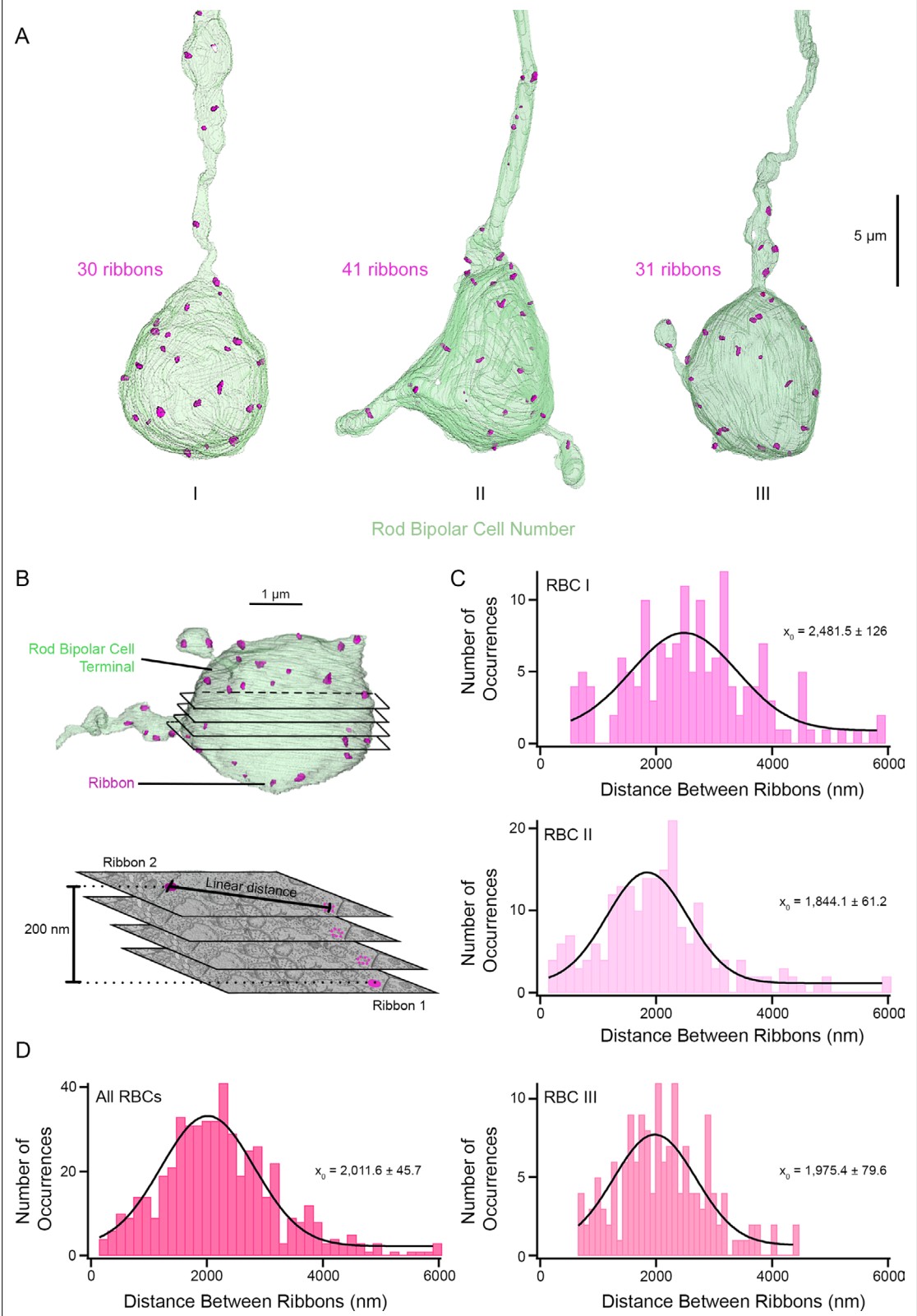

**Figure 9.** Distribution of measured distance between synaptic ribbons in three zebrafish RBCs. (**A**) Reconstruction of three RBC terminals closest to the ganglion cell layer resembling the shape and size of the mammalian RBC1. The ribbons are shown in magenta. Note that the total number of ribbons includes the 'floating' ribbons detached from the plasma membrane. RBCI and RBC II contained 1 and 8 floating ribbons, respectively. Scale bar, 5 µm. RBCs, rod bipolar cells. (**B**) Overview of the distance measurements. Top. 3D rendering of a single RBC terminal is shown in light green with

*Figure 9 continued on next page*

*Figure 9 continued*

ribbons in magenta. Black lines show an example of the different serial block face scanning electron microscopy (SBF-SEM) layers that are cut to obtain individual images. Bottom. Four sample SBF-SEM layers are shown with two example ribbons (magenta) located near each other but in different layers. If two ribbons were on the same layer, their linear distance was taken, whereas if they were located in different layers, their linear distance and height difference were used to calculate their actual distance using the Pythagorean Theorem. Each SBF-SEM layer has a thickness of 50 nm. A 3D volume movie of the synaptic terminal of a zebrafish bipolar neuron synaptic ribbon distribution and measurement is provided in *Video 1*.(C) Histograms showing the ribbon distance for the three RBCs. $x_0$ shows the mean of the distribution. (D) Histogram of the distances between each ribbon and its five nearest ribbons for all ribbons contained in the three RBCs. $x_0$ shows the mean of the distribution. Please note that the y-axis for **B** and **C** have different sizes, given that they represent the number of occurrences of each event.

## Discussion

$Ca^{2+}$ signaling plays a central role in regulating neurotransmitter release throughout the retina, but the spatial and temporal organization of these signals differs markedly between the outer and inner retinal circuits. In the outer retina, photoreceptors rely on graded, sustained calcium influx through L-type $Ca^{2+}$ channels at ribbon synapses to drive tonic glutamate release. These signals are relatively well characterized: the structure–function relationships between ribbon geometry, $Ca^{2+}$ channel distribution, and synaptic vesicle release have been extensively studied, and recent high-resolution studies *Grabner et al., 2022* have revealed nanometer-scale alignment between Cav channels and release sites. Postsynaptically, the identity of the glutamate receptor (e.g. mGluR6, KAR, AMPAR) and synaptic cleft geometry further tune the flow of visual information. In contrast, calcium dynamics in the inner retina, especially at bipolar cell terminals, are less well understood. The diversity of bipolar cell types, their complex patterns of ribbon organization, and the presence of both graded and spiking output modes add to this complexity. Moreover, presynaptic $Ca^{2+}$ signals in the inner retina are likely to establish distinct micro- or nanodomain mechanisms depending on ribbon structure, local channel density, and the spatial coupling between $Ca^{2+}$ influx and release sensors. Despite numerous studies that have focused on establishing retinal bipolar cells $Ca^{2+}$ signaling controlling NTR (*Graydon et al., 2011*; *Mennerick and Matthews, 1996*; *Singer and Diamond, 2003*; *Zenisek et al., 2003*; *Heidelberger et al., 1994*; *Palmer, 2010*), there remains significant uncertainty about how presynaptic $Ca^{2+}$ signals are organized across individual ribbon synapses, especially in small or non-classical synapses where standard imaging lacks the resolution to resolve sub-ribbon-scale features. Our study addresses this gap by applying a new immobile $Ca^{2+}$-ribbon indicator, enabling high-resolution tracking of $Ca^{2+}$ domains at the level of individual ribbons. This approach offers a window into nanodomain calcium signaling, allowing us to directly link structural heterogeneity to functional diversity in vesicle release. By capturing the fine-scale dynamics that govern synaptic output in the inner retina, this approach opens the door to a more complete understanding of how visual signals are shaped at their earliest synaptic relays.

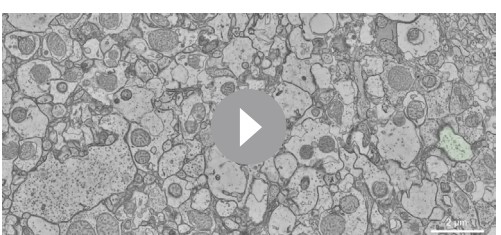

**Video 1.** 3D volume movie of the synaptic terminal of a zebrafish rod bipolar cell terminal showing the distribution of included synaptic ribbons and measurements. Rod bipolar cell (RBC) terminal is shown in light green with ribbons in magenta. Illustrations show how the measurements were obtained for ribbons on the same vs. different layers.

https://elifesciences.org/articles/105875/figures#video1

We note that a large correlation between the inferred ribbon size and active zone size would be expected if the size heterogeneity is partially due to the differences in the degree of labeling by the two ribbon-binding peptides. Therefore, the observed significant but moderate degree of correlation ($r=0.32$ or $r=0.52$) between the active zone area and the ribbon length or width provides an additional indication of large heterogeneity in the calcium influx between ribbons of similar size, increasing further the heterogeneity in neurotransmitter release by different ribbons.

## A deep look at the active zone $Ca^{2+}$ microdomain in RBC terminals

In this study, we developed a quantitative nanophysiology approach using ribeye-bound $Ca^{2+}$ indicators and fluorescently labeled RBP, combined with ratiometric dual-color imaging, to

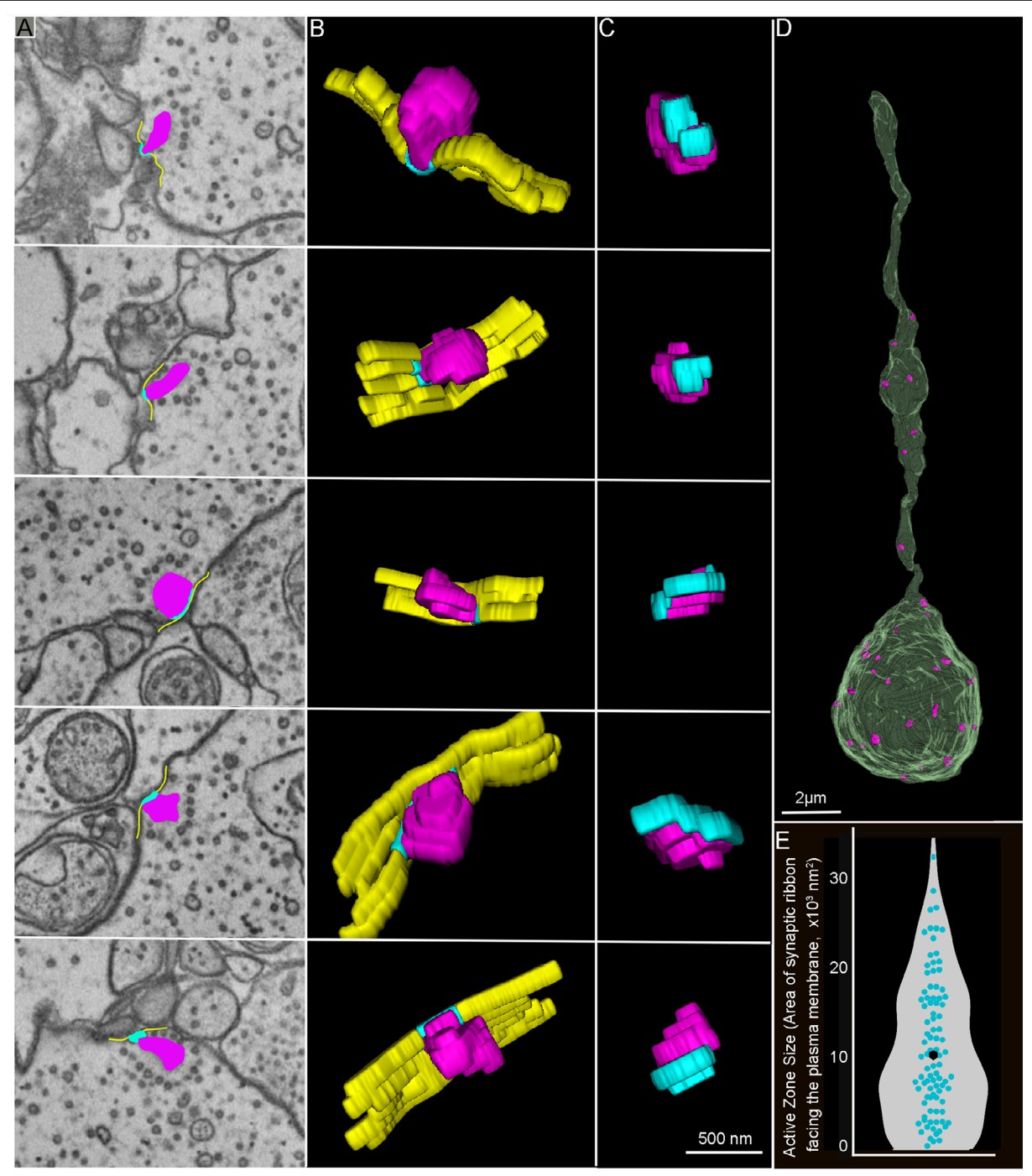

**Figure 10.** Serial block-face scanning electron microscopy analysis reveals heterogenous rod bipolar cell (RBC) ribbon shape, size, and area of the ribbon facing the plasma membrane. (**A–C**) Electron microscopy (EM) images of zebrafish RBC ribbon structures (**A**) and their respective 3D reconstructions to illustrate different shapes and sizes of synaptic ribbons (**A**, **B**, and **C**, magenta), plasma membrane (**A** and **B**, yellow), and the area of the ribbon facing the plasma membrane (**A**, **B**, and **C**, cyan). Each of the five rows of images illustrates one ribbon synapse from a zebrafish retinal RBC. A 3D reconstruction of the RBC synaptic terminal and ribbon from serial block face scanning electron microscopy (SBF-SEM) stacks is provided in *Video 2*. (**D**) 3D reconstruction of the RBC terminal closest to the ganglion cell layer resembles the shape and the size of the mammalian RBC1 (*Hellevik et al., 2024*). The ribbons are colored magenta. (**E**) Summary of the active zone size, the area of the ribbon associated with the plasma membrane measured in serial sections across the three RBCs from *Figure 9A*. The individual distribution of the three RBCs active zone sizes is provided in *Figure 10—figure supplement 1*. The z-size of the SBF-SEM sections is 50 nm, and each ribbon spans 2–5 consecutive sections. The solid cyan circles in the violin plots show individual synaptic ribbon measurements from three RBCs, with the average measurements shown in the solid black circle. Note that floating ribbons are not shown since they were not attached to the plasma membrane.

*Figure 10 continued on next page*

*Figure 10 continued*

The online version of this article includes the following figure supplement(s) for figure 10:

**Figure supplement 1.** Three rod bipolar cells (RBCs) active zone reconstructed from serial block-face scanning electron microscopy.

measure changes in $Ca^{2+}$ concentration at and along the ribbon axis. By targeting $Ca^{2+}$ indicators to the ribbon combined with fluorescently labeled RBP, we determined the $Ca^{2+}$ concentration along the ribbon at different distances from active zone and found that near the ribbon base, the $Ca^{2+}$ concentration could increase to an average of 26 µM upon depolarization. Given the localization of our indicator on the ribbon and minimal sensitivity to exogenous buffers, we believe that our ribbon-proximal signal represents $Ca^{2+}$ concentrations on the ribbon in locations very near release sites, likely some of those that contribute to vesicle resupply. Given that typical resolution for $Ca^{2+}$ imaging—whether using confocal or even STED microscopy—is limited to ~100–300 nm, the novelty of this study lies in the ability of the new immobile $Ca^{2+}$-ribbon indicator to potentially capture signals on even finer spatial scales, approaching the realm of nanophysiology. Using 2D STED microscopy combined with fluorescence lifetime imaging, Moser's lab achieved ~50 nm lateral resolution, enabling precise visualization of $Ca^{2+}$ channel clusters at single active zones in mouse cochlear inner hair cells (*Neef et al., 2018*). Notably, the $[Ca^{2+}]_i$ levels obtained from our higher resolution approach using confocal microscopy are similar to those that were reported for measurements in hair cells using Stimulated Emission Depletion (STED) lifetime measurements, emphasizing that our immobile ribbon-bound $Ca^{2+}$ indicator similarly enables nanoscopic (~tens of nanometers) resolution of localized $Ca^{2+}$ signals—complementing STED-based approaches like *Neef et al., 2018*. Previous work on RBCs found slowing of recovery from paired-pulse depression by EGTA, suggesting that $Ca^{2+}$ levels at distal sites on the ribbon might be important for the recruitment of new vesicles following depletion of the RRP and UFRP (*Mennerick and Matthews, 1996*; *von Gersdorff and Matthews, 1997*). Here, we measured directly the effect of EGTA on the spread of $Ca^{2+}$ signals and found that the $Ca^{2+}$ signal at the distal half of the ribbon was twofold lower than that at the location proximal to the membrane under conditions of low $Ca^{2+}$ buffering. It should be noted, however, that diffraction of light during the optical imaging procedure is likely to underestimate spatiotemporal $[Ca^{2+}]$ along the ribbon. Therefore, we used our measurement of the $Ca^{2+}$ current per ribbon to develop a detailed computational model that allowed us to resolve the expected profile of $Ca^{2+}$ microdomains governing UFRP fusion. The model results shown in *Figure 7* reveal, on a finer spatial scale, the distribution of $Ca^{2+}$ around a ribbon in the presence of various quantities of immobile endogenous buffer and added exogenous buffers. Our finding of high proximal $[Ca^{2+}]$ levels on the order of 26 µM recorded using a ribeye-bound low-affinity indicator dye is in qualitative agreement with the simulated high concentration of $Ca^{2+}$ in the microdomain at the base of the ribbon. Simulations also show that, as expected, the peak $Ca^{2+}$ microdomain amplitude is relatively insensitive to the concentration of endogenous immobile buffer and added EGTA, but its decay with distance is accelerated by increasing the concentrations of exogenous and endogenous buffers. In contrast, the addition of 2 mM BAPTA causes a profound reduction of the $Ca^{2+}$ microdomain amplitude and localizes the signal to a small volume around the ribbon. We also compared the impact of mobile vs immobile endogenous buffers on the $Ca^{2+}$ distribution, showing the much greater effect of mobile buffers on the microdomain amplitude and distance dependence. Although these results are generally expected (*Eisner et al., 2023*; *Neher, 1998*; *Matthews and Dietrich, 2015*), the simulation results estimate the $[Ca^{2+}]$ levels in the vicinity of the ribbon with greater spatial and temporal resolution. As expected, the ratios of $Ca^{2+}$ concentration values at the base of the ribbon vs. further away from the ribbon are much greater in

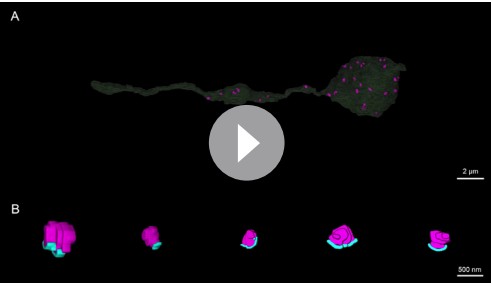

**Video 2.** A 3D reconstruction of the rod bipolar cell (RBC) synaptic terminal and ribbon from serial block face scanning electron microscopy (SBF-SEM) stacks. (A) 3D reconstruction of the RBC terminal (green) with included synaptic ribbons (colored magenta). (B) 3D reconstruction of the RBC synaptic ribbons shows different shapes and sizes of ribbon (magenta) and the AZ, the area of the ribbon facing the plasma membrane (cyan). AZ, active zone.

https://elifesciences.org/articles/105875/figures#video2

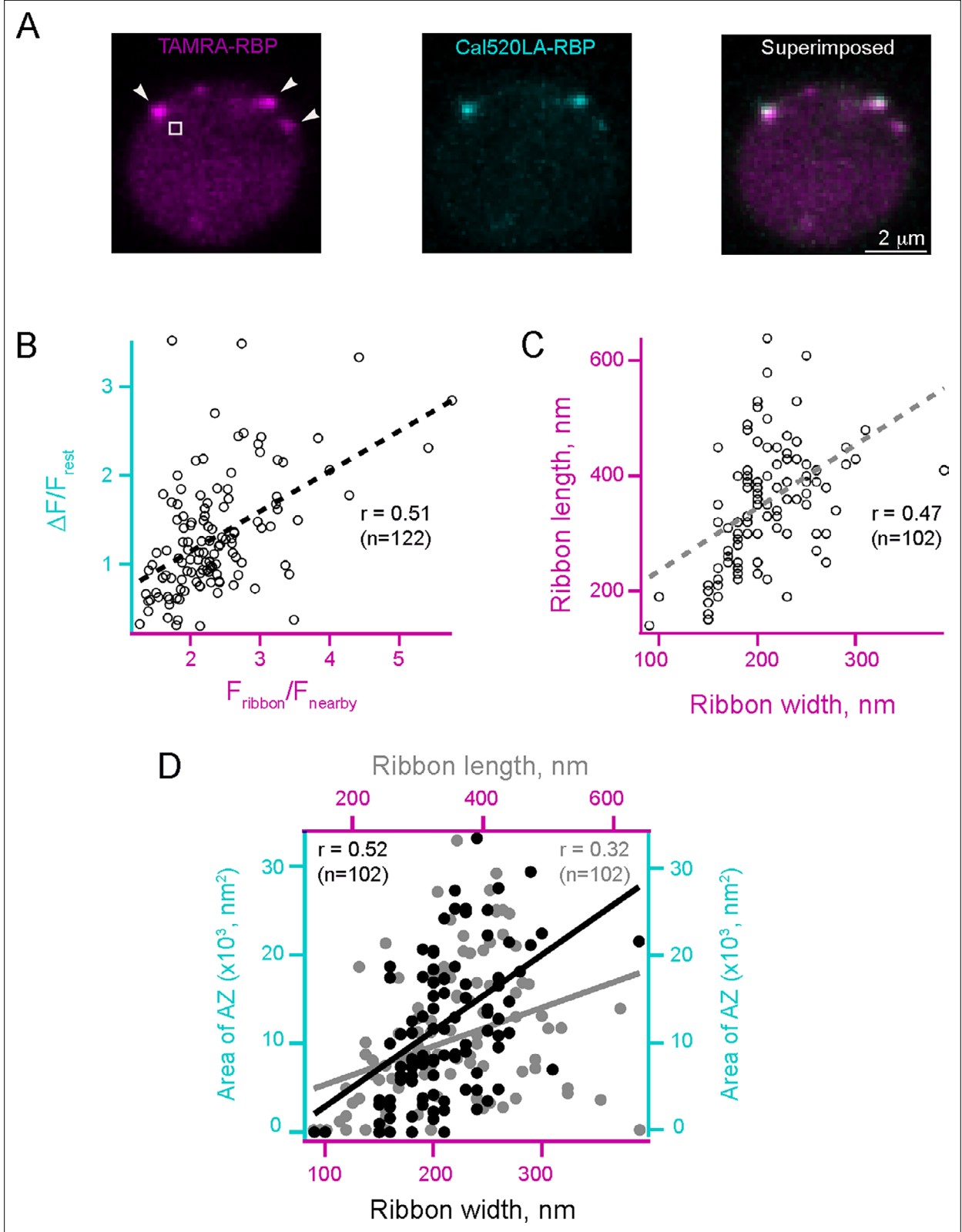

**Figure 11.** Heterogeneity of $Ca^{2+}$ microdomains in rod bipolar cell (RBC) terminals. Larger ribbons have stronger maximal $Ca^{2+}$ influx and larger CAZ. (**A**) Images show voltage-clamped RBC filled with a solution containing TAMRA-RBP to label ribbons before depolarization (left, magenta) and Cal520LA-RBP to measure the amplitude of $Ca^{2+}$ influx (middle, cyan) during depolarization and superimposed of the two (right) to compare the size of ribbon vs. maximal $Ca^{2+}$ influx. A time-lapse movie of the synaptic terminal of a zebrafish bipolar neuron during $Ca^{2+}$ influx is provided in *Video 3*. (**B**) Scatter plot

*Figure 11 continued on next page*

*Figure 11 continued*

of maximal Ca²⁺ influx (F/F_rest) vs. TAMRA-RBP. Dashed lines are linear regressions, and r is Pearson's correlation coefficient. (**C**) Scatter plot of maximal synaptic ribbon length vs. maximal synaptic ribbon width as estimated from SBF-SEM images. Dashed lines are linear regressions, and r is Pearson's correlation coefficient. (**D**) Scatter plot of area of AZ vs. ribbon width (filled black circles) and of area of AZ vs. ribbon length (filled gray circles). All the ribbons are included in the analysis. Dashed lines are linear regressions, and r is Pearson's correlation coefficient. AZ, Active Zone.

the model simulation than our experimentally determined proximal-to-distal Ca²⁺ ratio values due to the large size of the microscope PSF, which precludes precise localization of the corresponding Ca²⁺ signals. Furthermore, simulations show the effect of exogenous buffers with greater spatial resolution. However, the simulations fully agree with the overall increase of the proximal-to-distal Ca²⁺ concentration ratios by the application of exogenous Ca²⁺ buffers, especially BAPTA. Despite the technical challenges posed by the limited spatiotemporal resolution of optical imaging, the spatial resolution attained with the nanophysiology approach developed in our study is sufficient to differentiate Ca²⁺ signals associated with distal sites on the ribbon from those localized near the plasma membrane. Conceivably, the nanophysiology approaches established here could generally be applied to other classes of ribbon-containing cells, such as rods, cones, and hair cells.

Recent ultrastructural and functional studies have provided new insights into the nanoscale organization of calcium channels at ribbon synapses. In rod photoreceptors, Cav1.4 channels have been shown to localize not only at the base but also along both sides of the synaptic ribbon, in close proximity—within nanometers—to other active zone (AZ) proteins (*Grabner et al., 2022*). This spatial arrangement supports the idea that calcium influx is tightly restricted to subregions of the ribbon, enabling compartmentalized control of synaptic release. These findings are consistent with our own imaging results, which demonstrate localized, submicron Ca²⁺ signals that vary systematically with ribbon geometry. Together, these studies highlight a common theme: that nanodomain Ca²⁺ signaling at ribbon synapses is not only possible but structurally supported by the precise spatial alignment of Ca²⁺ channels and release machinery.

## Heterogeneity of Ca²⁺ microdomains across RBCs terminals

Our quantitative nanophysiology approach provides evidence for the heterogeneity in synaptic Ca²⁺ signals across zebrafish RBCs, suggesting variability in the Ca²⁺ microdomains as has been previously reported for hair cells (*Frank et al., 2009*; *Ohn et al., 2016*) and cone photoreceptors. Retinal bipolar neurons contain many ribbon active zones. This ranges from 45 to 60 in adult goldfish, as quantified by EM (*von Gersdorff et al., 1996*), or 25–45 in RBCs of adult living zebrafish, as estimated from laser scanning confocal microscopy (*Shrestha et al., 2023*). Our analyses of ribbon number across zebrafish RBCs through SBF-SEM arrived at an estimate of 31–40. The differences in the number of ribbons, as determined through confocal microscopy compared to SBF-SEM could be explained by the resolution limit encountered by the confocal image analyses. Our analyses of zebrafish RBC terminal SBF-SEM images and 3D projections provide the first evidence of diversity in synaptic ribbon shape and sizes across zebrafish RBC terminals. Furthermore, our quantitative measurements of SBF-SEM images reveal that synaptic ribbon width and length correlate with the active zone size, consistent with previous findings in hair cell ribbon synapses (*Martinez-Dunst et al., 1997*; *Wong et al., 2014*) and cone photoreceptor ribbon synapses (*Grassmeyer and Thoreson, 2017*). Our observation that maximal active zone Ca²⁺ influx correlates with ribeye fluorescence in live imaging is consistent with findings

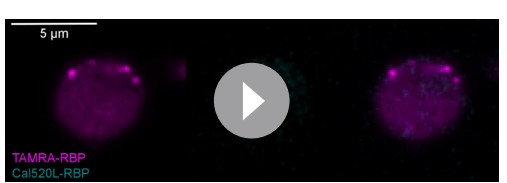

**Video 3.** A time-lapse movie of the synaptic terminal of a zebrafish bipolar neuron during Ca²⁺ influx. Left: Rod bipolar cell (RBC) terminal labeled with TAMRA-RBP shows the locations of synaptic ribbons (magenta spots). Middle: RBC terminal filled with Cal520LA-RBP shows the Ca²⁺ influx in 0.02 s, looped two times to see Ca²⁺ influx in 0.07 s. Right: Superimposed TAMRA-RBP and Cal520LA-RBP shows the Ca²⁺ influx as spot-like maxima near the membrane during depolarization. TAMRA-RBP 10 images before depolarization, and stacks of Cal520LA-RBP five images before depolarization and five images during depolarization were compiled to demonstrate the locations and magnitude of the Ca²⁺ influx. The interval between frames is 407 ms, total duration of the video 0.9 s. Each frame is an individual, unaveraged image.
https://elifesciences.org/articles/105875/figures#video3

previously reported in hair cells (*Ohn et al., 2016*). Thus, we expect larger active zones, with more $Ca^{2+}$ influx, to have more Cav channel expression, as demonstrated for hair cell ribbon synapses (*Ohn et al., 2016*). Heterogeneity among ribbon active zones has been previously reported in hair cells, where synaptic $Ca^{2+}$ microdomains varied substantially in amplitude and voltage-dependence within individual inner hair cells (*Frank et al., 2009*; *Ohn et al., 2016*). Studies in aquatic tiger salamander cone photoreceptors revealed that the amplitude and midpoint activation voltage of $Ca^{2+}$ signals varied across individual ribbons within the same cone. Additionally, local $Ca^{2+}$ signal dynamics at cone photoreceptor ribbons were found to be independently regulated (*Grassmeyer and Thoreson, 2017*). However, in comparing our findings with studies of ribbon size heterogeneity in hair cells, we note that we do not observe any systematic variation of ribbon size with larger-scale morphological features like the cell position, as compared to the reported interesting tonotopic variation of ribbon size with position along the cochlea (*Frank et al., 2009*; *Ohn et al., 2016*).

The faster, smaller, and more spatially confined $Ca^{2+}$ signals that are insensitive to the application of high concentrations of exogenous $Ca^{2+}$ buffers, referred to here as ribbon proximal $Ca^{2+}$ signals, could be due to $Ca^{2+}$ influx through Cav channel clusters beneath the synaptic ribbon (*Vaithianathan and Matthews, 2014*; *Zenisek et al., 2004*; *Beaumont et al., 2005*; *Berntson and Morgans, 2003*; *Issa and Hudspeth, 1994*; *Issa and Hudspeth, 1996*; *Llobet et al., 2003*; *Lv et al., 2012*; *Morgans et al., 2001*; *Neef et al., 2018*; *Raviola and Raviola, 1982*; *Roberts et al., 1990*; *Rodriguez-Contreras and Yamoah, 2001*; *Thoreson et al., 2013*; *Tucker and Fettiplace, 1995*; *Zenisek et al., 2003*). However, the variability in $Ca^{2+}$ signals at the distal ribbon, away from the plasma membrane, may result from the spread between two closely spaced microdomains; thus, if proximal $Ca^{2+}$ signals are variable, the distal $Ca^{2+}$ signals are also variable between ribbons. Alternatively, the variability in distal $Ca^{2+}$ signals could be due to additional mechanisms, for example, $Ca^{2+}$ influx from internal stores. Future studies should aim to elucidate the specific mechanisms underlying the observed heterogeneity in distal $Ca^{2+}$ signals. For instance, exploring differences in $Ca^{2+}$ release from intracellular stores or variations in $Ca^{2+}$ sequestration, as reported in goldfish, could reveal key contributing factors (*Kobayashi et al., 1995*). Although the potential impact of these intracellular $Ca^{2+}$ stores on $Ca^{2+}$ microdomain heterogeneity in hair cells has not been reported (*Frank et al., 2009*) previous studies in goldfish bipolar cell terminals have documented spatially restricted $Ca^{2+}$ oscillations in voltage-clamped retinal bipolar cells, occurring independently of membrane potential (*Francis et al., 2011*). To our knowledge, the sources of $Ca^{2+}$ for these oscillations remain unknown. However, previous studies in goldfish bipolar cells suggest that the endoplasmic reticulum (*Kobayashi et al., 1995*) and mitochondria (*Zenisek and Matthews, 2000*) may act as internal $Ca^{2+}$ stores. Notably, $IP_3$ receptors have been localized to retinal bipolar cell terminals in some species (*Micci and Christensen, 1996*; *Koulen et al., 2005*).

In the current study, we investigated possible mechanisms for heterogeneity in proximal $Ca^{2+}$ signals by measuring the ribbon size, particularly the area of the ribbon adjacent to the plasma membrane, where Cav channel clusters are located. Our SBF-SEM shows substantial variability in synaptic ribbon size, shape, and the area of the ribbon facing the plasma membrane where Cav channel clusters are tethered. Since both $Ca^{2+}$ signals and the area of the synaptic ribbon facing the plasma membrane are heterogeneous, we propose that the larger ribbons may anchor more channels, leading to larger $Ca^{2+}$ microdomains. Since local $Ca^{2+}$ signals control kinetically distinct neurotransmitter release components, heterogeneity in local $Ca^{2+}$ signals may alter the rate of vesicle release and allow them to function independently. Indeed, heterogeneity in neurotransmitter release kinetics has been proposed for the observed diversity in excitatory postsynaptic current amplitudes and kinetics reported from paired recordings of goldfish bipolar cells (*Palmer, 2010*) and ON-type mixed RBCs (*Kim and von Gersdorff, 2016*). The heterogeneity in the RBC $Ca^{2+}$ microdomains, synaptic ribbon shape, size, and

active zone area reported in this study may contribute to regulating the dynamic range of RBC ribbon synapses (*Euler et al., 2014*; *Trexler et al., 2005*; *Ke et al., 2014*) – a hypothesis that needs to be tested in future studies.

# Methods

**Key resources table**

| Reagent type (species) or resource | Designation | Source or reference | Identifiers | Additional information |
|---|---|---|---|---|
| Strain, strain background (*Danio rerio*) | WIK | ZIRC | ZDB-GENO-010531–2 | |
| Other | Hyaluronidase, type V | Sigma | H6254 | Enzyme (1100 units/mL) |
| Chemical compound, drug | NaCl | Fisher Scientific | S271-3 | 120 mM |
| Chemical compound, drug | KCl | Fisher Scientific | P217 | 2.5 mM |
| Chemical compound, drug | $CaCl_2$ | Honeywell / Fluka | 21117 | 0.5 mM |
| Chemical compound, drug | $MgCl_2$ | Fisher Scientific | AM95305 | 1 mM/3 mM |
| Chemical compound, drug | Glucose | Sigma | G5146 | 10 mM |
| Other | HEPES | J.T. Baker | 4018–04 | Buffer |
| Chemical compound, drug | DL-cysteine | Fluka | 30197 | 2 mM |
| Other | Papain | Fluka | 76220 | Enzyme (20–30 units/mL) |
| Other | Fire-polished glass Pasteur pipette | Fisher Scientific | 03-678-20A | fire-polished glass triturates |
| Other | RBP- Tetramethylrhodamine (TAMRA) | LifeTein LLC, NJ | GIDEEKPVDLTAGRRAG | Peptide |
| Other | Cal520HA-free High-affinity | AAT Bioquest | 21140 | Calcium indicator, Kd 320 nM |
| Other | Cal520LA-free Low-affinity | AAT Bioquest | 20642 | Calcium indicator, Kd ~90 μM |
| Other | Cal520HA-RBP | LifeTein | $Ca^{2+}$ indicator: 20610 RBP: $NH_2$-CIEDEEKPVDLTAGRRAC-COOH | Peptide Kd 320 nM |
| Other | Cal520LA-RBP | AAT Bioquest | $Ca^{2+}$ indicator: 20611 RBP: $NH_2$-CIEDEEKPVDLTAGRRAC-COOH | Peptide Kd ~90 μM |
| Chemical compound, drug | Cs-gluconate | Hello Bio, Incorporation | HB4822-10g | 120 mM |
| Chemical compound, drug | tetraethyl-ammonium-Cl | Tocris | 3068 | 10 mM |
| Chemical compound, drug | N-methyl-d-glucamine-EGTA | NMDG, Sigma EGTA, EMD Millipore | M2004 324626 | 0.2 mM |
| Chemical compound, drug | $Na_2ATP$ | Fisher Scientific | BP413 25 | 2 mM |
| Chemical compound, drug | $Na_2GTP$ | Fisher Scientific | 10106399001 | 0.5 mM |
| Chemical compound, drug | Ames' medium | US Biological Life Sciences | A1372-25 | buffer |

*Continued on next page*

*Continued*

| Reagent type (species) or resource | Designation | Source or reference | Identifiers | Additional information |
|---|---|---|---|---|
| Other | Glutaraldehyde 4% in 0.1 M Sodium-cacodylate buffer | Electron Microscopy | 16539–06 | buffer |
| Other | Olympus laser-scanning confocal microscope | Olympus, Shinjuku, Tokyo, Japan | Model IX 83 motorized inverted FV3000RS | microscope |
| Other | Electron microscopy system | Carl Zeiss Microscopy GmbH Jena, Germany | Zeiss 3-View | microscope |
| Software, algorithm | PatchMaster | HEKA Instruments, Inc, Holliston, MA | Version v2x90.4 | |
| Software, algorithm | Fluoview FV31S-SW Software | Olympus, Center Valley, PA | Version 2.3.1.163 | |
| Software, algorithm | Fiji/ImageJ, | https://imagej.nih.gov/ | Version 2.16.0/1.54 P | |
| Software, algorithm | Igor Pro Software | Wavemetrics, Portland, OR | Version 9.05 | |
| Software, algorithm | Microsoft Excel | Microsoft | Version 16.81 | |
| Software, algorithm | R Studio | R Studio | Version 2023.09.0+463 | |
| Software, algorithm | Adobe Photoshop | Adobe | Version 25.9 | |
| Software, algorithm | TrakEM plugin of ImageJ | https://imagej.nih.gov/ | Version 1.5 h | |

## Rearing of zebrafish

Male and female zebrafish (*Danio rerio*; 16–20 months) were raised under a 14 hr light/10 hr dark cycle and housed according to NIH guidelines and the University of Tennessee Health Science Center (UTHSC) Guidelines for Animals in Research. All procedures were approved by the UTHSC Institutional Animal Care and Use Committee (IACUC; protocol # 23–0459).

## Isolation of zebrafish retinal RBCs

Dissociation of RBCs was performed using established procedures (*Shrestha and Vaithianathan, 2022*). Briefly, retinas were dissected from zebrafish eyes and incubated in hyaluronidase (1100 units/ml) for 20 min. The tissue was washed with a saline solution containing 120 mM NaCl, 2.5 mM KCl, 0.5 mM $CaCl_2$, 1 mM $MgCl_2$, 10 mM glucose, and 10 mM HEPES, pH = 7.4 before being cut into quadrants. Each quadrant was incubated at room temperature for 25–40 min in the same saline solution, to which was added DL-cysteine and papain (20–30 units/ml; Sigma Millipore, St. Louis, MO) and triturated using a fire-polished glass Pasteur pipette. Individual cells were transferred to glass-bottomed dishes, allowed to attach for 30 min, and washed with saline solution before being used for experiments.

## Ribeye binding peptides

As a means of localizing the ribbons, custom peptides containing the ribbon binding sequence fused to TAMRA (tetramethylrhodamine; TAMRA: GIDEEKPVDLTAGRRAG) dye were synthesized, purified, and purchased from LifeTein (>95% purity, LifeTein LLC, NJ).

## $Ca^{2+}$ indicators

### Free $Ca^{2+}$ indicators

The potassium salts of high and low-affinity $Ca^{2+}$ indicators Cal-520 (high-affinity, $K_D$ 320 nM) and Cal-520N AM (low-affinity, $K_D$ 90 μM), referred to as Cal520HA and Cal520LA, respectively, were purchased from AAT Bioquest.

## Direct conjugation of Ca²⁺ indicator to cysteine-containing ribbon-binding peptides

To target Ca²⁺ indicators to the ribbon, custom-made cysteine-containing ribeye binding peptides NH₂-CIEDEEKPVDLTAGRRAC-COOH were synthesized and purchased from LifeTein to directly fuse with the fluorogenic 520 maleimide (purchased from AAT Bioquest) for high-affinity (HA) and low-affinity (LA Ca²⁺ indicator dyes). Each peptide at one mM concentration was mixed and incubated with two mM Cal-520 maleimide (20 mM stock solution in DMSO purchased from AAT Bioquest) for 1 hr at room temperature, then overnight at 4°C. Calibration of Ca²⁺ indicator dyes was performed as described (*Vaithianathan and Matthews, 2014*) and as detailed below. The conjugated Ca²⁺ indicators were stored at –20°C in smaller aliquots, each sufficient for a single day's experiments.

## Measurement of dissociation constants (K$_D$) for Ca²⁺ indicator peptides

The effective $K_D$ ($K_{eff}$) was obtained by measuring the fluorescence of Cal520HA-RBP and TAMRA-RBP in buffered Ca²⁺ solutions, determining the ratio between them (*Vaithianathan and Matthews, 2014*), and using the Grynkiewicz equation to determine the Ca²⁺ concentration [Ca²⁺] from this ratio (*Grynkiewicz et al., 1985*). However, this was not possible for the low-affinity indicator Cal520LA due to the large Ca²⁺ levels required to calibrate it. Thus, $K_{eff}$ for Cal520LA-RBP could be larger than the $K_{1/2}$ provided by the manufacturer for Cal520LA ($K_D$, 90 µM), as reported previously with $K_{eff}$, measurements of OB-5N in inner hair cells (*Neef et al., 2018*). Thus, our estimates of local Ca²⁺ concentrations obtained using Cal520LA represent the lower bounds of the underlying true values. As noted in the Results section, the same may be true for the high-affinity Cal520HA despite its accurate $K_{eff}$ estimate, due to potential dye saturation effects.

## RBC voltage clamp recording

Whole-cell patch-clamp recordings were made from isolated RBCs, as described previously (*Vaithianathan and Matthews, 2014*; *Vaithianathan et al., 2016*; *Shrestha et al., 2022*). Briefly, a patch pipette containing pipette solution (120 mM Cs-gluconate, 10 mM tetraethyl-ammonium-Cl, 3 mM MgCl₂, 0.2 mM *N*-methyl-d-glucamine-EGTA, 2 mM Na₂ATP, 0.5 mM Na₂GTP, 20 mM HEPES, pH = 7.4) was placed on the synaptic terminal, as described previously. The patch pipette solution also contained a fluorescently-labeled RBP peptide (TAMRA-RBP) to mark the positions of the ribbons and either (1) free Ca²⁺ indicators Cal520HA-free (*Figures 1 and 2*) and Cal520LA-free (*Figures 2, 3 and 5*) to demonstrate our nanophysiological approach or (2) ribeye-bound Ca²⁺ indicators Cal520HA-RBP (*Figure 3* and *Figure 5—figure supplement 1* and *Figure 8—figure supplement 1*) and Cal520LA-RBP (*Figures 3–5 and 8*) to measure local ribbon-associated Ca²⁺ signals. Current responses from the cell membrane were recorded under a voltage clamp with a holding potential (V$_H$) of –65 mV that was stepped to 0 mV (t$_0$) for 10 milliseconds. These responses were recorded with a patch clamp amplifier running PatchMaster software (version v2x90.4; HEKA Instruments, Inc, Holliston, MA). Membrane capacitance, series conductance, and membrane conductance were measured via the sine DC lock-in extension in PatchMaster and a 1600 Hz sinusoidal stimulus with a peak-to-peak amplitude of 10 mV centered on the holding potential (*Vaithianathan et al., 2013*).

## Acquisition of confocal images

Confocal images were acquired using an Olympus model IX 83 motorized inverted FV3000RS laser-scanning confocal microscopy system (Olympus, Shinjuku, Tokyo, Japan) running FluoView FV31S-SW software (Version 2.3.1.163; Olympus, Center Valley, PA) equipped with a 60 X silicon objective (NA 1.3), all diode laser combiner with five laser lines (405, 488, 515, 561, & 640 nm), a true spectral detection system, a hybrid galvanometer, and a resonant scanning unit. Fluorescently labeled ribeye binding peptide (RBP) *Zenisek et al., 2004* and Ca²⁺ indicator were delivered to RBC via a whole-cell patch pipette placed directly at the cell terminal. We waited for 30 s after break-in to allow Cal520HA to reach equilibrium with the patch-pipette before obtaining the first fluorescence image. Rapid x-t line scans at the ribbon location were performed to localize synaptic ribbons (*Figure 1Aii*) and to monitor local changes in Ca²⁺ concentration at a single ribbon, as we demonstrated previously, to estimate the Ca²⁺ levels at the plasma membrane, and to track a single synaptic vesicle at ribbon locations (*Vaithianathan and Matthews, 2014*; *Francis et al., 2011*). The z-projection from a series of confocal optical sections through the synaptic terminal (*Figure 1Ai*) illustrates ribbon labeling (magenta spots).

RBP fluorescence was used to localize a synaptic ribbon and to define a region for placing a scan line perpendicular to the plasma membrane, extending from the extracellular space to the cytoplasmic region beyond the ribbon to monitor changes in the $Ca^{2+}$ concentration along the ribbon axis (*Figure 1Aii*). The focal plane of the TAMRA-RBP signal was carefully adjusted for sharp focus to avoid potential errors arising from the high curvature near the top of the terminal and the plane of membrane adherence to the glass coverslip at the bottom of the terminal. Sequential dual laser scanning was performed at rates of 1.51 milliseconds per line. Two-color laser scanning methods allowed observation of $Ca^{2+}$ signals (*Figure 1C*, **cyan**) throughout the full extent of the ribbon in voltage-clamped synaptic terminals, while the ribbon and cell border were imaged with a second fluorescent label. The exchange of TTL (transistor-transistor logic) pulses between the patch-clamp and imaging computers synchronized the acquisition of electrophysiological and imaging data. The precise timing of imaging relative to voltage-clamp stimuli was established using PatchMaster software to digitize horizontal-scan synced pulses from the imaging computer in parallel with the electrophysiological data (*Figure 1B*). Acquisition parameters, such as pinhole diameter, laser power, PMT gain, scan speed, optical zoom, offset, and step size were kept constant between experiments. Sequential line scans were acquired at 1–2 millisecond/line and 10 μs/pixel with a scan size of 256×256 pixels.

Bleed-through between the channels was confirmed with both lasers using the imaging parameters we typically use for experiments. To test bleed-through from the RBP channel (TAMRA-RBP) to the $Ca^{2+}$ indicator (Cal-520HA and Cal-520LA), whole-cell recordings from RBC terminals were performed with patch pipette solution that contained TAMRA-RBP or the aforementioned $Ca^{2+}$ indicators, and line-scan images were collected and analyzed using the same procedures used for experimental samples.

## Point spread function

The lateral and axial point spread function (PSF) was obtained as described previously (*Vaithianathan and Matthews, 2014*; *Vaithianathan et al., 2019*; *Shrestha and Vaithianathan, 2022*). Briefly, an XYZ scan was performed through a single 27 nm bead and the maximum projection in the xy-plane was fit to the Gaussian function using Igor Pro software. We obtained the full width at half maximum (FWHM) values for x and y-width of 268 and 273 nm, respectively, in the lateral (x-y plane) and for y-z width of 561 nm in the axial (y-z axis) resolution.

## Photobleaching

We minimized photobleaching and phototoxicity during live-cell scanning by using fast scan speed (10 microseconds/pixel), low laser intensity (0.01–0.06% of maximum), and low pixel density (frame size, 256 × 256 pixels). We estimated photobleaching using x-t line scans of Cal520HA or Cal520LA and TAMRA-RBP in the absence of stimulation, with the same imaging parameters used for experimental samples.

## Data analysis

Quantitative FluoView x-t and x-y scans were analyzed initially with ImageJ software (https://imagej.nih.gov/) and subsequently with Igor Pro software (Wavemetrics, Portland OR) for curve fitting and production of the figures. Data from PatchMaster software were initially exported to Microsoft Excel (Version 16.81) for normalizing and averaging and exported from MS Excel to Igor Pro (Version 9.05) for curve fitting and production of the l figures.

## Analysis of x-t scan data
### X-axis profile
To determine the $Ca^{2+}$ signals along the ribbon axis, we spatially averaged the x-axis profile intensity of RBP (i.e. a horizontal row of pixels, see *Figure 1C*, *top*) to determine the position of the center of the ribbon and estimate the location of the plasma membrane. The parameter $x_0$ is the peak of the Gaussian fit, giving the x-position of the center of the ribbon. The x-axis profile is also used to obtain the spatial profile of $Ca^{2+}$ signals before, during, and after with respect to the ribbon profile. To identify the $Ca^{2+}$ signals specific to the ribbon location, we fit x-axis intensity profiles with the equation $f(x) = s(x) + g(x)$, where $s(x)$ is a sigmoid function that describes the transition from intracellular to extracellular background fluorescence at the edge of the cell, given by

$s\left(x\right) = b - c\left(1 - exp\left(\left(x_{1/2} - x\right)/d\right)\right)$, and $g\left(x\right)$ is a Gaussian function that represents the fluorescence of RBP, given by $g\left(x\right) = aexp\left(-\left(x - x0\right)^2/w^2\right)$, as described (**Vaithianathan et al., 2016**). The parameters $x_{1/2}$ and $x_0$ were taken as the x-axis positions of the plasma membrane and the fluorescence emitter, respectively. While parameter b is intracellular background fluorescence, *c* is extracellular background fluorescence, *d* is the slope of the sigmoid, *a* is the peak amplitude of emitter fluorescence, and *w* is $\sqrt{2}$ times the standard deviation of the Gaussian function, in practice, the latter parameters were highly constrained by the data or by the measured PSF, leaving only $x_{1/2}$ and $x_0$ as free parameters in the fitting. **Figure 1D** demonstrates that the peak of the Ca$^{2+}$ signals ($x_0$, cyan; **Figure 1D**) during the stimuli is proximal to the ribbon center ($x_0$, magenta; **Figure 1D**) towards the plasma membrane ($x_{1/2}$, magenta; **Figure 1D**), as expected for Ca$^{2+}$ influx originating from Ca$^{2+}$ channels localized in the plasma membrane (**Roberts, 1994**; **Naraghi and Neher, 1997**; **Zucker and Fogelson, 1986**).

## T-axis profile

The temporal profiles of the Cal520, Cal520-RBP, and TAMRA-RBP signals were determined by analyzing the time-axis profile of the x-t line scan to obtain the kinetics of the Ca$^{2+}$ transient with respect to the ribbon. We determined the baseline kinetics by averaging the fluorescence obtained immediately before depolarization. The timing of depolarization and the amplitude of the Ca$^{2+}$ current was obtained from the PatchMaster software. The rising phase of the Ca$^{2+}$ transient was fit with the sigmoid function, the peak of which is referred to as the peak amplitude Ca$^{2+}$ transient.

We used this baseline profile of ribbon-proximal Ca$^{2+}$ signals to distinguish signals proximal or distal to the ribbon, despite the distance between these two signals being within the PSF. For example, the ribbon-proximal signals were obtained by averaging five pixels of the temporal profile of the Ca$^{2+}$ signals (obtained with Cal520 or Cal520-RBP) between the $x_{1/2}$ and $x_0$ values obtained in the x-axis profile of TAMRA-RBP. The distal profile was obtained similarly but was 5 pixels after $x_0$ towards the cytoplasm.

## Quantifying the kinetics of the Ca$^{2+}$ transients

The decay phase of the fluorescence transients in **Figures 2 and 3** were fit using the standard bi-exponential sum with time constants $\tau_{fast}$ and $\tau_{slow}$:

$$f_{decay}\left(t\right) = F_{peak}\left[a_{fast}\exp\left(-\frac{t - t_{peak}}{\tau_{fast}}\right) + a_{slow}\exp\left(-\frac{t - t_{peak}}{\tau_{slow}}\right)\right]$$

Where $F_{peak}$ and $t_{peak}$ are the peak fluorescence and the time at which this peak has been reached, and $a_{fast}$ and $a_{slow}$ are the relative proportions of the two decay components, specified as percentages and satisfying $a_{fast} + a_{slow} = 100\%$. The decay parameters were found using a global differential evolution optimization algorithm (**Storn and Price, 1997**), repeating the optimization 400 times starting with different initial parameter values, to ensure true global minimum of the fit error is reached. We note, however, that bi-exponential data fitting is known to be an ill-conditioned procedure, highly sensitive to noise and other recording characteristics such as the duration of the measurement. Therefore, much higher signal-to-noise ratio would be required for reliable uncertainty quantification of the corresponding kinetic fit parameters, so we only fit the trial-averages data for fluorescence traces in **Figures 2 and 3** as a rough description of temporal decay of the signal; no precise quantitative conclusions should be drawn from the decay time values. Still, time-course fit of the averaged traces may provide a qualitative-level comparison of stimulus-evoked fluorescence transients recorded with different dyes at different locations with respect to the ribbon.

## Analysis of x-y scan data

We analyzed the rate of loading the TAMRA-RBP and Ca$^{2+}$ indicator into the RBC terminal with ImageJ software by placing a square region of interest (ROI; 5 × 5 μm) and using a Plot z-axis profile function to obtain spatially averaged TAMRA-RBP and Ca$^{2+}$ indicator fluorescence as a function of time. The rising phase of TAMRA-RBP and Ca$^{2+}$ indicator fluorescence was obtained by fitting the rising to the

peak to the single exponential function using the curve fitting function in Igor Pro 9 software. The rate of the exponential function is defined as the rate of fluorescence loading to the terminal.

For analysis comparing the fluorescence intensity of ribbons and $Ca^{2+}$ influx elicited by 100 ms stimuli (*Figure 11A & B*), 10 images of both TAMRA-RBP and Cal520LA-RBP were collected in sequence, followed by imaging Cal520LA-RBP only during 200 ms depolarization, and the sequence ended with 10 images of both TAMRA-RBP and Cal520LA-RBP in sequence to confirm the ribbon locations. Two-color imaging were obtained sequentially with a frame interval of 407 ms, and Cal520LA-RBP only during 100 ms stimuli was 205 ms. Images before and after were then averaged to obtain the ribbon location. Synaptic ribbon fluorescence was quantified as the ratio of TAMRA fluorescence to the fluorescence of the nearby RBC cytoplasm, measured approximately eight to nine pixels away ($F_{ribbon}/F_{nearby}$), measuring the pixel with the highest intensity. The change in Cal520LA-RBP fluorescence, used as a proxy for active zone $Ca^{2+}$ influx, was estimated as $\Delta F/F_{rest}$, where $F_{rest}$ is the fluorescence intensity before depolarization at −65 mV, and $\Delta F$ is the difference when depolarized to 0 mV. $Ca^{2+}$ indicator intensity was calculated as the average fluorescence of nine pixels centered on the pixel with the greatest fluorescence increase.

## Experimental design and statistical methods

We did not use any statistical methods to determine the sample size prior to experiments. Mean value variances were reported as ± the standard error of the mean (SEM). The statistical significance of differences in average amplitudes of $Ca^{2+}$ current, capacitance, synaptic ribbon size and number, and $Ca^{2+}$ transients were assessed using unpaired, two-tailed t-tests with unequal variance using R Studio (Version 2023.09.0+463) and Igor Pro software.

### Amplitude analysis for *Figure 8*

Amplitude data were processed and analyzed using R Studio software (Version 2023.09.0+463). The data were divided into two groups, that of individual cells and of multiple cells, and analyzed as follows. Multiple individual cells from each condition (proximal 10 mM EGTA Cal520LA-RBP and distal 10 mM EGTA Cal520LA-RBP) were analyzed to compare the variability between the $Ca^{2+}$ amplitude measurements of the ribbons from each cell. Having verified the normality of the data to ensure the validity of the statistical test, we used one-way ANOVA and Tukey's honest significant difference test to assess the specific statistical differences between ribbons. To compare the variability between the amplitude measurements of different cells, we compared readings for various cells. If there were multiple measurements for a single ribbon in a specific cell, those readings were averaged. In this case, the normality of the data was again verified and Welch's ANOVA was employed to assess the between-group differences, as it does not assume homogeneity of variance, with the Games-Howell post-hoc test being used to understand the specific differences between cells.

## Computational modeling of $Ca^{2+}$ dynamics

Spatio-temporal $Ca^{2+}$ dynamics is simulated in a volume shown in *Figure 6*, which is a 3D box of dimensions (1.28×1.28 ×1.1) $\mu m^3$. Assuming an approximately spherical bouton of diameter 5 μm and an average of 36 ribbons per terminal, one obtains synaptic volume per ribbon of 1.81 $\mu m^3$, which matches the volume of our box domain. Ribbon serves as an obstacle to $Ca^{2+}$ and buffer diffusion, and has an ellipsoidal shape with semi-axis of 190 nm along the Y- and Z-axes (for a total height of 380 nm), and a semi-axis of 70 nm along the X-axis. It is attached to the plasma membrane by a thin ridge (arciform density) of dimensions 60×30×30 nm. $Ca^{2+}$ ions enter through four channels (or clusters of channels) at the base of the ribbon indicated by black disks in *Figure 6*, forming a square with a side length of 80 nm; this highly simplified channel arrangement is sufficient to capture the level of detail in spatial $[Ca^{2+}]$ distribution that we seek to resolve. Each of the four clusters admits a quarter of the total $Ca^{2+}$ current. $Ca^{2+}$ current values and current pulse durations are listed in *Figure 7*. Although the actual distribution of Cav channels is more heterogeneous, channels close to the ribbon play a greater role in neurotransmitter release (*Moser et al., 2020*).

The simulation includes a single dominant $Ca^{2+}$ buffer with molecules possessing a single $Ca^{2+}$ ion binding site. We assume that only immobile buffer is present in a patch-clamped cell, since any mobile buffer would diffuse into recording pipette. Simulations with mobile buffer in an unpatched cell are also provided (*Figure 6—figure supplement 1B & C*). Buffer parameter values were set to values

reported by *Burrone et al., 2002*, namely dissociation constant of 2 μM, and total concentration of 1.44 mM (total buffering capacity at rest of 720). We also examine the impact of the assumption on buffer concentration by repeating the simulation with a lower buffering capacity of 100 (total buffer concentration of 200 μM). The $Ca^{2+}$ ion diffusivity is set to 0.22 $\mu m^2$/ms (*Allbritton et al., 1992*).

To minimize the number of undetermined parameters, we assumed the same parameters for the $Ca^{2+}$ clearance on all surfaces, with one high-capacity but lower affinity process simulating $Na^+/Ca^{2+}$ exchangers, and one lower capacity but higher affinity process simulating $Ca^{2+}$ extrusion by ATPase PMCA and SERCA pumps. We used the Hill coefficient of $m=2$ for the $Ca^{2+}$ sensitivity of the SERCA pumps on all surfaces except the bottom surface (*Lytton et al., 1992*), whereas the Hill coefficient of $m=1$ corresponds to PMCA pumps on the bottom surface (*Brini, 2009*). Thus, the flux of $[Ca^{2+}]$ across the surface (ions extruded per unit area per unit time) is described by

$$\Phi_{Ca} = D_C \left( \mathbf{n} \cdot \nabla C \right) = \frac{A_{NCX} C}{K_{NCX} + C} + \frac{A_P C^m}{K_P^m + C^m} - \Phi_{Leak},$$

where $C$ is the $Ca^{2+}$ concentration, $D_C$ is the $Ca^{2+}$ diffusivity, and $\mathbf{n} \cdot \nabla C$ is the gradient of $[Ca^{2+}]$ normal to the boundary. The constant leak term $\Phi_{Leak}$ ensures that the flux is zero at the resting value $[Ca^{2+}]_{rest}$ = $C_0$=0.1 μM. The $Ca^{2+}$ clearance rate and affinity parameters for the pumps and exchangers are listed in *Supplementary file 3*. We note, however, that the density of $Na^+$-$Ca^{2+}$ exchangers and PMCA/SERCA pumps are rough estimates and have a significant effect on results only for longer stimulation durations of over 100–200 ms. Reaction-diffusion equations for $[Ca^{2+}]$ and buffer are solved numerically using Calcium Calculator software version 6.10.7 (*Matveev et al., 2002*; *Matveev, 2023*).

## SBF-SEM

The eyecups of adult (16–20-month-old) zebrafish were dissected in bicarbonate-buffered Ames' medium and, after removing the lens, the eyecups were fixed in 4% glutaraldehyde in 0.1 M sodium-cacodylate buffer at RT for 1 hr, followed by overnight fixation at 4°C. Thereafter, samples were treated to generate blocks for serial block face scanning electron microscopy (SBF-SEM) according to the published protocol (*Della Santina et al., 2016*), in this case using Zeiss 3-View SBF-SEM. Multiple 35 μm montages were acquired at a 5 nm X-Y resolution and a 50 nm Z resolution across the entire retina from the outer (photoreceptor) layer to the inner (ganglion) cell layer. Images were inspected visually and structures of interest were traced using the TrakEM plug-in in ImageJ software (NIH/Fiji), as described below.

### Analysis of serial block-face scanning electron microscopy images

SBF-SEM image stacks of the zebrafish retina were imported, aligned, visualized, and analyzed using the TrakEM plugin (version 1.5 hr) in ImageJ software (NIH). RBC1 were identified based on their characteristic morphology and 4–7 μm terminal size, as previously described (*Hellevik et al., 2023*). RBCs, synaptic ribbons, pre-synaptic membranes, and the area of the ribbon touching the plasma membrane were traced, painted using the TrakEM brush features, and rendered in 3D for structural visualization. The area of the synaptic ribbon touching the RBC membrane was quantified using the measuring tools in ImageJ and images were prepared using Adobe Photoshop software (version 25.9). The distance between each ribbon from three different RBCs and its nearest five ribbons (*Figure 9B* and *Video 1*) was measured as follows: Ribbons on the same plane had their linear distance measured, whereas the distance between ribbons that were close to each other but in different layers was calculated using the Pythagorean Theorem.

## Acknowledgements

This research was funded by the National Institutes of Health (NIH) National Eye Institute (NEI), award numbers R01EY030863, 3R01EY030863-02S1, and 3R01EY030863-03S1, and a UTHSC College of Medicine Faculty Research Growth Award (to TV), R01 EY032396, P30EY026878, R01NS122388 (DZ), and a University of Wisconsin-Madison (UW2020) WARF discovery award (to MH) which funded the acquisition of the 3D view serial block-face electron microscope. We would like to acknowledge the UW Madison School of Medicine electron microscopy facility for processing and imaging samples and thank Dr. Alex Dopico, the Van Vleet Chair of Excellence and Professor in the Department of

Pharmacology, Addiction Science, and Toxicology, UTHSC, for critical reading, and Dr. Kyle Johnson Moore in the UTHSC Office of Research for editing the manuscript.

## Additional information

### Funding

| Funder | Grant reference number | Author |
|---|---|---|
| National Eye Institute | R01EY030863 | Thirumalini Vaithianathan |
| National Eye Institute | 3R01EY030863-02S1 | Thirumalini Vaithianathan |
| National Eye Institute | 3R01EY030863-03S1 | Thirumalini Vaithianathan |
| National Eye Institute | R01 EY032396 | David Zenisek |
| National Eye Institute | P30EY026878 | David Zenisek |
| National Eye Institute | R01NS122388 | David Zenisek |

The funders had no role in study design, data collection and interpretation, or the decision to submit the work for publication.

### Author contributions

Nirujan Rameshkumar, Formal analysis, Investigation, Writing – original draft, Writing – review and editing; Abhishek P Shrestha, Formal analysis, Writing – review and editing; Johane M Boff, Formal analysis, Visualization, Writing – original draft, Writing – review and editing; Mrinalini Hoon, Resources, Formal analysis, Funding acquisition, Methodology, Writing – review and editing; Victor Matveev, Conceptualization, Formal analysis, Methodology, Writing – review and editing; David Zenisek, Conceptualization, Resources, Funding acquisition, Writing – review and editing; Thirumalini Vaithianathan, Conceptualization, Resources, Data curation, Software, Formal analysis, Supervision, Funding acquisition, Validation, Investigation, Visualization, Methodology, Writing – original draft, Project administration, Writing – review and editing

### Author ORCIDs

Nirujan Rameshkumar ⓘ https://orcid.org/0009-0006-7848-3390
Abhishek P Shrestha ⓘ https://orcid.org/0000-0003-1576-8584
Victor Matveev ⓘ https://orcid.org/0000-0002-9867-5190
David Zenisek ⓘ https://orcid.org/0000-0001-6052-0348
Thirumalini Vaithianathan ⓘ https://orcid.org/0000-0002-7686-5284

### Ethics

Male and female zebrafish (Danio rerio; 16 ~20 months) were raised under a 14 h light/10 h dark cycle and housed according to NIH guidelines and the University of Tennessee Health Science Center (UTHSC) Guidelines for Animals in Research. All procedures were approved by the UTHSC Institutional Animal Care and Use Committee (IACUC; protocol # 23-0459).

Reviewer #1 (Public Review): https://doi.org/10.7554/eLife.105875.4.sa1
Reviewer #2 (Public review): https://doi.org/10.7554/eLife.105875.4.sa2
Reviewer #3 (Public review): https://doi.org/10.7554/eLife.105875.4.sa3
Author response https://doi.org/10.7554/eLife.105875.4.sa4

## Additional files

### Supplementary files

Supplementary file 1. Effect of exogenous $Ca^{2+}$ chelators alter $Ca^{2+}$ signals gradient along synaptic ribbon measured with Cal520LA-RBP. There were significant differences between proximal vs distal measured as $\Delta F/F_{rest}$, in all conditions, as found through paired-sample t-test analysis performed on RStudio. Differences were smaller between proximal vs distal $Ca^{2+}$ signals in 0.2 mM, 2 mM, and

10 mM EGTA conditions, but more prominent with 2 mM BAPTA (0.2 mM EGTA: proximal vs distal: $p=0.0027$, 2 mM EGTA: proximal vs distal: $p=0.034$, 10 mM EGTA: proximal vs distal $p=0.00013$, 2 mM BAPTA: proximal vs. distal: $p=0.0073$, n=22).

Supplementary file 2. Effect of exogenous $Ca^{2+}$ chelators alters the $Ca^{2+}$ signal gradient along the synaptic ribbon measured with Cal520H-RBP. There were significant differences between proximal vs distal measured as $\Delta F/F_{rest}$, in all conditions as found through paired-sample t-test analysis performed on RStudio. Differences were smaller between proximal vs distal $Ca^{2+}$ signals in 0.2 mM EGTA and 2 mM EGTA conditions, but more prominent with 10 mM EGTA, and further enhanced with 2 mM BAPTA (0.2 mM EGTA: proximal vs distal $p=0.00135$, n=19; 2 mM EGTA: proximal vs distal $p=7.4\cdot10^{-4}$, n=23; 10 mM EGTA: proximal vs distal $p=1.4\cdot10^{-5}$, n=29; 2 mM BAPTA: proximal vs distal $p=0.0046$, n=22).

Supplementary file 3. Model parameters for $Ca^{2+}$ diffusion, buffering, and clearance. Simulations were performed assuming an endogenous buffer with a total concentration of either 1.4 mM total resting buffering capacity of 720 *Oesch and Diamond, 2011*; *Burrone et al., 2002*; *Coggins and Zenisek, 2009*, or a lower concentration of 200 µM corresponding to a buffering capacity of 100. Simulations in *Figure 7* assumes immobile endogenous buffer, while this file assumes a typical value of buffer mobility of 0.05 µm²/ms. The $Ca^{2+}$ clearance parameters are adapted from *Graydon et al., 2011*; *Jarsky et al., 2010*; *Mennerick and Matthews, 1996*; *Singer and Diamond, 2003*; *Snellman et al., 2009*; *Von Gersdorff and Mathews, 1994*; *Augustine et al., 1991*. Note that flux units of (µM µm)/ms are equivalent to $10^{-21}$ mol/(µm²ms)=602 ions/(µm²ms). Properties of EGTA and BAPTA (not listed here) are summarized in *Burrone et al., 2002*.

MDAR checklist

## Data availability

Source data have been provided for all figures and figure supplements (https://doi.org/10.5061/dryad.qfttdz0vh). The code for the analyses presented in this paper is openly accessible at https://github.com/mvvik/Zebrafish-Calcium-Microdomains (copy archived at *Matveev, 2025*) and https://github.com/vaithianathanlab/calcium_microdomains_project (copy archived at *Vaithianathan Lab, 2025*).

The following dataset was generated:

| Author(s) | Year | Dataset title | Dataset URL | Database and Identifier |
|---|---|---|---|---|
| Thirumalini V, Nirujan R, Abhishek S, Johane B, Mrinalini H, Victor M, David Z | 2025 | Data from: Nanophysiology approach reveals diversity in calcium microdomains across Zebrafish retinal bipolar ribbon synapses | https://doi.org/10.5061/dryad.qfttdz0vh | Dryad Digital Repository, 10.5061/dryad.qfttdz0vh |

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
