## [Editor Report · eLife Assessment]

The study introduces new tools for measuring the intracellular calcium concentration close to transmitter release sites, which may be relevant for synaptic vesicle fusion and replenishment. This approach yields **important** new information about the spatial and temporal profile of calcium concentrations near the site of entry at the plasma membrane. This experimental work is complemented by a coherent, open-source, computational model that successfully describes changes in calcium domains. The conclusions are **solid** and well supported by the data.

---

## [Referee Report · Reviewer #1 (Public Review)]

This paper describes technically impressive measurements of calcium signals near synaptic ribbons in zebrafish bipolar cells. The data presented provides high spatial and temporal resolution information about calcium concentrations along the ribbon at various distances from the site of entry at the plasma membrane. This is important information. The experiments appear to be well-done and provide strong evidence for the main conclusions reached.

Strengths

The technical aspects of the measurements are impressive. The authors use calcium indicators bound to the ribbon and high-speed line scans to resolve changes with a spatial resolution of ~250 nm and temporal resolution of less than 10 ms. These spatial and temporal scales are much closer to those relevant for vesicle release than previous measurements. Hence the results provide a unique window onto these events.

The use of calcium indicators with very different affinities and of different intracellular calcium buffers helps provide confirmation of key results.

---

## [Referee Report · Reviewer #2 (Public review)]

Summary:

The study introduces new tools for measuring intracellular Ca2+ concentration gradients around retinal rod bipolar cell (rbc) synaptic ribbons. This is done by comparing the Ca2+ profiles measured with mobile Ca2+ indicator dyes versus ribbon-tethered (immobile) Ca2+ indicator dyes. The Ca2+ imaging results provide a straightforward demonstration of Ca2+ gradients around the ribbon and validate their experimental strategy. This experimental work is complemented by a coherent, open-source, computational model that successfully describes changes in Ca2+ domains as a function of Ca2+ buffering. In addition, the authors try to demonstrate that there is heterogeneity among synaptic ribbons within an individual rbc terminal.

Strengths:

The study introduces a new set of tools for estimating Ca2+ concentration gradients at ribbon AZs, and the experimental results are accompanied by an open-source, computational model that nicely describes Ca2+ buffering at the rbc synaptic ribbon. In addition, the dissociated retinal preparation remains a valuable approach for studying ribbon synapses. Lastly, excellent EM.

Comments on revisions:

Several concerns were raised about the kinetic analyses, and the authors have carefully acknowledged the critiques. The ideal outcome would have been a more complete kinetic readout and analyses (in particular a better readout of risetime would have improved the results). In the absence of a suitable readout of the risetime, the authors scaled back their claims and improved on the description of the falling phase of the signals. The authors have given a reasonable response under the circumstances.

In addition, the authors provided more context to their results.

I have no further concerns.

---

## [Referee Report · Reviewer #3 (Public review)]

Summary:

In this study, the authors have developed a new Ca indicator conjugated to the peptide, which likely recognizes synaptic ribbons and have measured microdomain Ca near synaptic ribbons at retinal bipolar cells. This interesting approach allows one to measure Ca close to transmitter release sites, which may be relevant for synaptic vesicle fusion and replenishment. Though microdomain Ca at the active zone of ribbon synapses has been measured by Hudspeth and Moser, the new study uses the peptide recognizing synaptic ribbons, potentially measuring the Ca concentration relatively proximal to the release sites.

Strengths:

The study is, in principle, technically well done, and the peptide approach is technically interesting, which allows one to image Ca near the particular protein complexes. The approach is potentially applicable to other types of imaging.

Weaknesses:

Peptides may not be entirely specific, and genetic approach tagging particular active zone proteins with fluorescent Ca indicator proteins may well be more specific. The readers should be aware of this, when interpreting the results.

---

## [Author Response]

The following is the authors’ response to the previous reviews

**Reviewer #1 (Public review):**
This paper describes technically-impressive measurements of calcium signals near synaptic ribbons in goldfish bipolar cells. The data presented provides high spatial and temporal resolution information about calcium concentrations along the ribbon at various distances from the site of entry at the plasma membrane. This is important information. Important gaps in the data presented mean that the evidence for the main conclusions is currently inadequate.StrengthsThe technical aspects of the measurements are impressive. The authors use calcium indicators bound to the ribbon and high speed line scans to resolve changes with a spatial resolution of ~250 nm and temporal resolution of less than 10 ms. These spatial and temporal scales are much closer to those relevant for vesicle release than previous measurements.The use of calcium indicators with very different affinities and of different intracellular calcium buffers helps provide confirmation of key results.

Thank you very much for this positive evaluation of our work.

WeaknessesMultiple key points of the paper lack a statistical test or summary data from populations of cells. For example, the text states that the proximal and distal calcium kinetics in Figure 2A differ. This is not clear from the inset to Figure 2A - where the traces look like scaled versions of each other. Values for time to half-maximal peak fluorescence are given for one example cell but no statistics or summary are provided. Figure 8 shows examples from one cell with no summary data. This issue comes up in other places as well.

Thank you for this fair and valuable feedback. Following also the suggestion by the Editor, we have now removed the rise-time kinetic fitting results from the manuscript and only retain the bi-exponential decay time constant values. Further, we explicitly detail the issues with kinetic fitting, and state that the precise quantitative conclusions should not be drawn from the differences in kinetic parameters (pages 7 and 2728).

We have included the results of paired-t-tests to compare the amplitudes of proximal vs. distal calcium signals shown in Fig. 2A & B, Fig. 3C & D, Fig. 4C & D, Fig. 5A-D, and Fig. 8E&F. Because proximal and distal calcium signals were obtained from the same ribbons within 500-nm distances, as the Reviewer pointed out, “the traces look like scaled versions of each other”. For experiments where we make comparisons across cells or different calcium indicators, as shown in Fig. 3E & F, Fig.5E, and Fig. 8B&C, we have included the results of an unpaired t-test. We have also included the t-test statistics information in the respective figure legends in the revised version.

In Figure 8, we have shown example fluorescence traces from two different cells at the bottom of the A panel, and example traces from different ribbons of RBC a in the D, and the summary data is described in B-C and E-F, with statistics provided in the figure legends.

The rise time measurements in Figure 2 are very different for low and high affinity indicators, but no explanation is given for this difference. Similarly, the measurements of peak calcium concentration in Figure 4 are very different with the two indicators. That might suggest that the high affinity indicator is strongly saturated, which raises concerns about whether that is impacting the kinetic measurements.

Yes, we do believe that the high-affinity indicator is partially saturated, and therefore, the measurement with the low-affinity indicator dye is a more accurate reflection of the measured Ca^2+^ signal. We now state this more explicitly in the text. Further, we note that the rise time values are no longer listed due to lack of statistical significance for such comparisons, as noted above.

**Reviewer #2 (Public review):**
Summary:The study introduces new tools for measuring intracellular Ca2+ concentration gradients around retinal rod bipolar cell (rbc) synaptic ribbons. This is done by comparing the Ca2+ profiles measured with mobile Ca2+ indicator dyes versus ribbon-tethered (immobile) Ca2+ indicator dyes. The Ca2+ imaging results provide a straightforward demonstration of Ca2+ gradients around the ribbon and validate their experimental strategy. This experimental work is complemented by a coherent, open-source, computational model that successfully describes changes in Ca2+ domains as a function of Ca2+ buffering. In addition, the authors try to demonstrate that there is heterogeneity among synaptic ribbons within an individual rbc terminal.Strengths:The study introduces a new set of tools for estimating Ca2+ concentration gradients at ribbon AZs, and the experimental results are accompanied by an open-source, computational model that nicely describes Ca2+ buffering at the rbc synaptic ribbon. In addition, the dissociated retinal preparation remains a valuable approach for studying ribbon synapses. Lastly, excellent EM.

Thank you very much for this positive evaluation of our work.

Comments on revisions:Specific minor comments:(1) Rewrite the final sentence of the Abstract. It is difficult to understand.

Thank you for pointing that out. We have updated the final sentence of the Abstract.

(2) Add a definition in the Introduction (and revisit in the Discussion) that delineates between micro- and nano-domain. A practical approach would be to round up and round down. If you round up from 0.6 um, then it is microdomain which means ~ 1 um or higher. Likewise, round down from 0.3 um to nanodomain? If you are using confocal, or even STED, the resolution for Ca imaging will be in the 100 to 300 nm range. The point of your study is that your new immobile Ca2-ribbon indicator may actually be operating on a tens of nm scale: nanophysiology. The Results are clearly written in a way that acknowledges this point but maybe make such a "definition" comment in the intro/discussion in order to: (1) demonstrate the power of the new Ca2+ indicator to resolve signals at the base of the ribbon (effectively nano), and (2) (Discussion) to acknowledge that some are achieving nanoscopic resolution (50 to 100nm?) with light microscopy (as you ref'd Neef et al., 2018 Nat Comm).

Thank you for the valuable comments. We have now provided this information in the introduction and discussion.

(3) Suggested reference: Grabner et al. 2022 (Sci Adv, Supp video 13, and Fig S5). Here rod Cav channels are shown to be expressed on both sides the ribbon, at its base, and they are within nanometers from other AZ proteins. This agrees with the conclusions from your imaging work.

Thank you for the valuable suggestion. We have now provided this information in the introduction and discussion.

(4) In the Discussion, add a little more context to what is known about synaptic transmission in the outer and inner retina.. First, state that the postsynaptic receptors (for example: mGluR6-OnBCs vs KARs-OffBCs, vs. AMPAR-HCs), and possibly the synaptic cleft (ground squirrel), are known to have a significant impact on signaling in the outer retina. In the inner retina, there are many more unknowns. For example, when I think of the pioneering Palmer JPhysio study, which you sight, I think of NMDAR vs AMPAR, and uncertainty in what type postsynaptic cell was patched (GC or AC....). Once you have informed the reader that the postsynapse is known to have a significant impact on signaling, then promote your experimental work that addresses presynaptic processes: "...the new tool and results allow us to explore release heterogeneity, ribbon by ribbon in dissociated preps, which we eventually plan to use at ribbon synapses within slices......to better understand how the presynapse shapes signaling......".

Thank you for the valuable comments. We have now provided this information in the introduction and discussion.

**Reviewer #3 (Public review):**
Summary:In this study, the authors have developed a new Ca indicator conjugated to the peptide, which likely recognizes synaptic ribbons and have measured microdomain Ca near synaptic ribbons at retinal bipolar cells. This interesting approach allows one to measure Ca close to transmitter release sites, which may be relevant for synaptic vesicle fusion and replenishment. Though microdomain Ca at the active zone of ribbon synapses has been measured by Hudspeth and Moser, the new study uses the peptide recognizing synaptic ribbons, potentially measuring the Ca concentration relatively proximal to the release sites.Strengths:The study is, in principle, technically well done, and the peptide approach is technically interesting, which allows one to image Ca near the particular protein complexes. The approach is potentially applicable to other types of imaging.

Thank you very much for this appreciation.

Weaknesses:Peptides may not be entirely specific, and genetic approach tagging particular active zone proteins with fluorescent Ca indicator proteins may well be more specific. Although the authors are aware of this and the peptide approach is generally used for ribbon synapses, the authors should be aware of this, when interpreting the results.

We acknowledge the reviewer’s point and believe the peptides and genetic approaches to measure local calcium signals have their merits, each with separate advantages and disadvantages.

**Reviewer #1 (Recommendations for the authors):**
The revisions helped with some concerns about the original paper, but some issues were not adequately addressed. I have left two primary concerns in my public review. To summarize those:The difference in kinetics of proximal and distal locations is emphasized and quantified in the paper, but the quantification consists of a fit to the average responses. This does not give an idea of whether the difference observed is significant or not. Without an estimate of the error across measurements the difference in kinetic quoted is not interpretable.

Thank you for this feedback. Since the kinetics information is a minor part of the manuscript, we have followed the Editor’s advice to significantly tone down the comparison of kinetic fit parameters (completely removing the rise-time comparisons), in order to put more focus on the better-documented conclusions. We also note that we did establish statistical significance of the differences in fluorescence signal amplitudes.

Somewhat relatedly, the difference in amplitude and kinetics of the calcium signals measured with low and high affinity indicators is quite concerning. The authors added one sentence stating that the high affinity indicator might be saturated. This is not adequate. Should we distrust the measurements using the high affinity indicator? The differences between the results using the low and high affinity indicators is in some cases large - e.g. larger than the differences cited as a key result between distal and proximal locations. This issue needs to be dealt with directly in the paper.Thank you for this feedback. Yes, the measurements from high-affinity indicators cannot report the Ca2+ as accurately as low-affinity indicators. However, the value of HA indicators is in their ability to detect lowamplitude signals that lower-affinity indicators may miss due to lower signal-to-noise resolution. We added a sentence on page 12 to further stress this point.Related to the point about statistics, it is not clear how to related the horizontal lines in Figure 8 to the actual measurements. It is critical for the evaluation of the conclusions from that figure to understand what is plotted and what the error bars are on the plotted data.

We apologize for the earlier ambiguity in Fig. 8. In this figure, we first compare proximal (panel B) and distal (panel C) calcium signals across several RBCs, labeled RBC-a through RBC-d. Each RBC contains multiple ribbons, and for each cell, we present the average calcium signals from multiple ribbons using box plots in panels B and C. In these box plots, the horizontal lines represent the average calcium signal for each cell, while the size of the error bars reflects the variability in proximal and distal calcium signals among the ribbons within that RBC.

For example, RBC-a had five identifiable ribbons. In panels D–F, we use RBC-a to illustrate the variability in calcium signals across individual ribbons. Specifically, we distinguished proximal and distal calcium signals from five ribbons (ribbons 1–5) within RBC-a. When feasible, we acquired multiple x–t line scans at a single ribbon, shown now as individual data points, to assess variability in calcium signals recorded from the same ribbon.

The box plots in panels E and F display the average calcium signal (horizontal lines) for each ribbon, based on multiple recordings. These plots demonstrate considerable variability between ribbons of RBC-a. Importantly, the lack of or minimal error bars for repeated measurements at the same ribbon indicates that the proximal and distal calcium signals are consistent within a ribbon. These findings emphasize that the observed variability among ribbons and among cells reflects true biological heterogeneity in local calcium domains, rather than experimental noise.